# SCUT : Spectral Clustering for Unsupervised classification Trees

## Abstract

Many real-world datasets lack reliable annotations, making it necessary to organize data in an unsupervised and interpretable manner. While clustering methods can reveal latent structure, they typically do not provide a principled mechanism for assigning new samples to existing clusters, and many hierarchical approaches suffer from limited scalability on large datasets. We propose Spectral Clustering for Unsupervised classification Trees (SCUT), a hierarchical clustering framework based on algebraic connectivity that enables consistent out-of-sample routing within the learned structure. By leveraging a feature-splitting approach, SCUT also supports *ante-hoc* interpretability of clustering decisions. Formally, SCUT recursively partitions the data by solving the Normalized Cut (NCUT) problem—a graph-partitioning formulation that seeks to split a graph into balanced subsets while minimizing the total connection strength between them—on a bipartite graph, producing structured and geometrically well-defined subspaces. From an algorithmic standpoint, SCUT is designed with scalability in mind. By exploiting truncated Singular Value Decomposition (SVD) computations on bipartite representations and an efficient recursive construction, it achieves a best-case time complexity of $\Omega(mn \log n)$ and a worst-case complexity of $O(mn^2)$. Moreover, a simple modification of the splitting rule guarantees $O(mn \log n)$ complexity in both the best and worst cases, strengthening its suitability for large-scale datasets. We demonstrate, both visually and quantitatively, that SCUT captures the intrinsic structure of data more effectively than existing methods, while offering competitive performance compared to common hierarchical clustering algorithms.

## 1 Introduction

Clustering is the task of grouping similar objects together. Given a set of data points (also called observations) and a pairwise distance (or similarity) measure, clustering consists of forming homogeneous groups of observations. In most clustering problems, data are unlabeled, making clustering primarily an unsupervised learning technique widely used to uncover the underlying structure of a dataset. This structure is often hierarchical: for example, in a collection of movies, films can be grouped by genre and further subdivided into subgenres. Hierarchical clustering algorithms aim to capture such multilevel organization within a single dendrogram.

Methods for hierarchical clustering are typically divided into two categories: agglomerative and divisive. Agglomerative approaches, such as Ward's method, proceed bottom-up by recursively merging clusters until only one remains. Divisive approaches operate top-down, recursively splitting clusters into subclusters. Let $m$ denote the number of features and $n$ the number of observations. While merging requires examining $\Theta(n^2)$ candidate pairs, splitting in its most general form involves $\Theta(2^n)$ possible bipartitions. In practice, agglomerative methods typically exhibit time complexity at least $\Theta(mn^2)$Müllner (2011), and often $\Theta(mn^3)$, which limits their applicability to large datasets.

A substantial body of work has investigated spectral clustering and Normalized Cut (NCUT)–based formulations. The seminal works of Shi & Malik (2000) and Weiss (1999) established relaxations of the NCUT objective via eigenvector computations, forming the foundation of modern spectral clustering. Subsequent

developments, including (Ng et al., 2001) and the tutorial of Von Luxburg (2007), further clarified the theoretical properties of graph Laplacians and their use in clustering through spectral embeddings. The notion of algebraic connectivity, introduced by Fiedler (1973), provides the theoretical basis for interpreting the second-smallest eigenvalue of the Laplacian as a measure of graph connectivity and underlies recursive spectral partitioning strategies. Dhillon (2001) and Zha et al. (2001) extended these ideas to bipartite graphs, showing that clustering documents and terms can be performed efficiently through singular value decomposition (SVD) without explicitly constructing the full similarity matrix. In these approaches, the dominant cost is the computation of a truncated SVD, which can be carried out in $O(mnk)$ time for $k$ singular vectors using modern randomized or Lanczos-based methods. While several works have proposed out-of-sample extensions for spectral embeddings, notably via Nyström-based approximations (e.g., Bengio et al. (2004)), these methods typically provide post-hoc embedding extensions rather than an intrinsic hierarchical routing mechanism. Subsequent research has explored hierarchical and divisive variants, including NCUT-based tree constructions and cluster trees for efficient nearest-neighbor search (e.g., Kushnir & Silwal (2025)), as well as applications to structured and textual data (e.g., Wieling & Nerbonne (2010); Zheng et al. (2018); Sugahara & Okamoto (2023); Ienco et al. (2009)). While these methods leverage spectral principles or hierarchical decompositions, they typically focus either on flat spectral partitions, task-specific adaptations, or efficiency-oriented tree structures, and generally lack a principled mechanism for inductive assignment of new samples within a hierarchical framework.

In this work, we propose SCUT (Spectral Clustering for Unsupervised classification Trees), a divisive hierarchical clustering framework based on algebraic connectivity. SCUT recursively approximates the NCUT objective on bipartite graphs using singular value decomposition, while retaining all original features at each recursive step. The resulting hierarchy supports both efficient construction and consistent out-of-sample routing.

This work makes the following contributions:

- We introduce SCUT, a divisive hierarchical clustering framework that approximates the Normalized Cut (NCUT) objective on bipartite graphs via singular value decomposition and leverages algebraic connectivity—the second-smallest eigenvalue of the graph Laplacian—to construct recursive linear splits while preserving the full feature space at each step.

- We propose a principled density-based thresholding strategy for split selection and show that the resulting recursive partitions induce strictly convex subspaces in the transformed space $T(I)$, where $T$ denotes the normalization mapping applied to the input space $I$, providing geometric guarantees on cluster structure.

- We design a native out-of-sample routing mechanism that enables consistent assignment of new observations within the learned hierarchy without recomputing the clustering.

- We provide a formal complexity analysis establishing a best-case time complexity of $\Omega(mn \log n)$ and a worst-case complexity of $O(mn^2)$, together with a simple modification that guarantees $\Theta(mn \log n)$ complexity in both cases.

Beyond its empirical performance, SCUT is particularly suited to large and evolving datasets, where scalability, geometric structure, and consistent inductive assignment are essential.

The article is organized as follows. In Section 2, we introduce the spectral background and present the SCUT methodology, including its hierarchical construction, inference mechanism, interpretability analysis, and computational complexity. Section 3 reports experimental results on diverse datasets. Section 4 discusses limitations and potential extensions, and Section 5 concludes.

## 2 Material and Methods

We begin by defining the notation and framework—previously introduced in several studies such as (Von Luxburg, 2007)—used to describe our novel method, SCUT.

## 2.1 Spectral clustering

Let $G = (V, E)$ be a weighted graph and $w$ a positive weight function for edges in G. The volume of $X$ is defined as

$$vol(X) = \sum_{u \in X, v \in V} w(u, v) \tag{1}$$

where $X$ is a subset of $V$. Intuitively, the volume measures the weight of the connections in $X$. If $V$ is partitioned into two disjoint sets $(X, Y)$, that is $X \cup Y = V$ and $X \cap Y = \emptyset$, the *cut* of the partition $(X, Y)$ is defined as

$$cut(X, Y) = \sum_{u \in X, v \in Y} w(u, v) \tag{2}$$

Intuitively, the cut of a partition measures the total weight of the connections between subsets $X$ and $Y$. Note that $cut(X, Y) = cut(Y, X)$ and $cut(X, Y) \leq vol(X)$.

In clustering, the goal is to partition a set into subsets that are as homogeneous as possible. In our case, this means identifying subsets that minimize the raw *cut* value. However, as noted in Shi & Malik (2000), minimizing the raw cut value alone often leads to highly unbalanced partitions. To address this issue, they introduced the notion of *normalised cut*:

$$Ncut(X, Y) = \frac{cut(X, Y)}{vol(X)} + \frac{cut(X, Y)}{vol(Y)} \tag{3}$$

Since the edges that define $cut(X, Y)$ are a subset of the edges defining $vol(X)$ and $vol(Y)$, it follows that $cut(X, Y) < vol(X)$ and $cut(X, Y) < vol(Y)$. Intuitively, minimizing $Ncut$ means to search for a nearly equally-sized partition $(X, Y)$ where $vol(X)$ and $vol(Y)$ are comparable, and $X$ and $Y$ are weakly connected.

Formally, using Theorem 3 from Dhillon (2001), the $Ncut(X, Y)$ objective function can be conveniently rewritten using the unnormalized graph Laplacian $D - A$ as:

$$Ncut(X, Y) = \frac{q^\mathsf{T}(D - A)q}{q^\mathsf{T} A q} \tag{4}$$

where $D$ is a $n \times n$ diagonal matrix with $D_{ii} = vol(\{u_i\})$ and $D_{ij} = 0$ for $i \neq j$, $A$ is a $n \times n$ weight matrix with $A_{ij} = w(u_i, v_j)$ if $i \neq j$ and $A_{ii} = 0$, and q is a partitioning vector representing $(X, Y)$:

$$q_i = \begin{cases} +\sqrt{\frac{vol(Y)}{vol(X)}} & \text{for } u_i \in X \\ -\sqrt{\frac{vol(X)}{vol(Y)}} & \text{for } u_i \in Y \end{cases} \tag{5}$$

The normalized cut problem becomes:

$$\min_q \frac{q^\mathsf{T}(D - A)q}{q^\mathsf{T} A q} \tag{6}$$

This is a discrete optimization problem as the entries of the partitioning vector $q$ are only allowed to take two particular values. Most importantly, this problem is NP-complete (Shi & Malik, 2000). Spectral graph partitioning is an effective heuristic to find solutions avoiding as much as possible local minima barriers (Donath & Hoffman, 1972; Shi & Malik, 2000; Ng et al., 2001). It discards the discreteness condition and instead it allows that the $q_i$'s take arbitrary values in $\mathbb{R}$. This leads to the relaxed optimisation problem

$$\min_{q \in \mathbb{R}^n} q^\mathsf{T}(D - A)q \qquad \text{subject to } q \perp \mathbb{1}, \| q \| = \sqrt{n} \tag{7}$$

By the Rayleigh-Ritz Theorem (Lütkepohl, 1997), it can be seen that the solution of this problem is given by the vector $q$ which is the eigenvector corresponding to the second smallest eigenvalue of $D - A$ (recall that the smallest eigenvalue of $D - A$ is 0 with eigenvector $\mathbb{1}$), also called Fiedler vector (Fiedler, 1973). So we can approximate a minimizer of $Ncut(X, Y)$ by the second eigenvector of the Laplacian $D - A$. However, in order to obtain a partition of the graph, we need to re-transform the real-valued solution vector $q$ of the relaxed problem into a discrete partitioning vector. This can be done by using the sign of $q$ as partition function, that is by chosing $X = \{u_i \in V, q_i \geq 0\}$ and $Y = \{u_i \in V, q_i < 0\}$.

## 2.2 Spectral Clustering with Bipartite Graphs

In many practical applications, data is represented in the form of a tabular matrix. Specifically, let $W$ be an $n \times m$ tabular matrix, where $n$ is the number of observations and $m$ the number of features. Typically, the number of observations exceeds the number of features ($n > m$). Applying clustering directly to the $n$ observations is equivalent to constructing a graph with $n$ nodes and $\frac{n(n-1)}{2}$ edges, which becomes computationally expensive when $n$ is large. An alternative is to model the data as a bipartite graph consisting of $n$ observation nodes and $m$ feature nodes, with edge weights defined by the entries of a matrix $W$. Although this approach may seem more costly at first—since the matrices $A$ and $D$ would be of size $(n+m) \times (n+m)$—the bipartite nature of the graph allows for a more efficient representation. In particular, matrix $A$ takes the form of a symmetric block matrix:

$$A = \begin{bmatrix} 0 & W \\ W^\mathsf{T} & 0 \end{bmatrix} \tag{8}$$

This structure enables several computational optimizations. In fact, it has been shown (Dhillon, 2001; Zha et al., 2001) that clustering the $n$ observations (and/or the $m$ features) can be performed without explicitly constructing the full matrix $A$; only the original matrix $W$ is needed for the computation. More formally, we consider a bipartite graph G = (U ∪ V, E), where $U \cap V = \emptyset$ and $\forall (a,b) \in E$, $(a \in U \wedge b \in V) \vee (a \in V \wedge b \in U)$, and let $n = |U|$, $m = |V|$. $G$ has weighted edges and it is represented by a matrix $W \in \mathbb{R}^{n \times m}$ where each entry $W_{i,j} = w(u_i, v_j)$, for $i = 1 \ldots n$ and $j = 1 \ldots m$, denotes the weight of the edge between node $u_i$, an observation, and node $v_j$, a feature. For bipartite graphs $G$, the diagonal matrix $D$ becomes the symmetric block matrix:

$$D = \begin{bmatrix} D_1 & 0 \\ 0 & D_2 \end{bmatrix} \tag{9}$$

where $D_1(i,i) = vol(\{u_i\})$ and $D_1(i,k) = 0$ for $i \neq k$, and $D_2[j,j] = vol(\{v_j\})$ and $D_2[k,j] = 0$ for $j \neq k$. In practice, only the diagonals of the matrices $D_1$ and $D_2$ are needed for the computations.

As shown (Dhillon, 2001; Zha et al., 2001), we can find the Fiedler vector $f$ of G by using a singular value decomposition (SVD) of $D_1^{-1/2} W D_2^{-1/2}$, where $D_h^{-1/2}[k,k]$ denotes $\sqrt{D_h[k,k]}$ for $h = 1, 2$. The Fiedler vector $f$ is the concatenation of the second largest left and right singular vectors, respectively denoted $f_u, f_v$. Based on them, the partition is defined as $X_U = \{u_i \in U, f_u \geq 0\}$ and $Y_U = \{u_i \in U, f_u < 0\}$, and $X_V = \{v_i \in V, f_v \geq 0\}$ and $Y_V = \{v_i \in V, f_v < 0\}$ (Dhillon, 2001; Zha et al., 2001). Namely, $f_u$ gives a bipartitioning of the observations and $f_v$ of the features.

## 2.3 The SCUT method

In this section, we introduce SCUT (Spectral Clustering for Unsupervised classification Trees), a novel method for hierarchical clustering of tabular data. SCUT consists of two main components: tree construction and inference. The tree is constructed by recursively applying a modified version of the spectral clustering algorithm proposed in (Dhillon, 2001; Zha et al., 2001), resulting in a dendrogram that hierarchically partitions the data. At each recursive step, the algorithm computes and stores a linear projection that fully characterizes the corresponding partition. During inference, these learned linear projections are used to assign new data points to appropriate leaves of the classification tree.

SCUT introduces several technical innovations that lead to important improvements over existing spectral clustering approaches:

- At each recursive step, SCUT clusters samples while retaining all original features, avoiding feature reduction.

- It interprets the Fiedler vector $f_u$ as samples from a one-dimensional distribution, which can be analyzed based on its density.

- Each split yields an explicit linear decision function derived from the spectral decomposition. By storing these functions at internal nodes, SCUT supports inductive assignment of new samples without recomputing the clustering (Algorithm 2).

- SCUT leverages a physical analogy to interpret $f_v$, offering *ante-hoc* explanations for clustering decisions (Zha et al., 2001).

- Complexity analysis shows that SCUT achieves a best-case time complexity of $\Omega(mn \log n)$ and a worst case complexity of $\mathcal{O}(mn^2)$, offering improvements over other methods.

As in (Dhillon, 2001; Zha et al., 2001), SCUT operates on non-negative tabular data. When real-valued features are present, simple preprocessing techniques such as min–max scaling, adding a constant offset, or sigmoid transformations can be used to obtain a non-negative representation without altering the relative structure of the data. Categorical variables can be handled through one-hot encoding.

### 2.3.1 Split characterisation from properties of the SVD

We use properties of SVD to learn a linear projection that characterizes a split. Let $W_N = D_1^{-1/2} W D_2^{-1/2}$ be the normalized matrix. Its SVD is given by:

$$W_N = U \Sigma V^\mathsf{T} \tag{10}$$

where $U$ and $V$ are the matrices whose columns are the left and right singular vectors, respectively, and $\Sigma$ is a diagonal matrix containing the singular values $\sigma = \text{diag}(\Sigma)$ in decreasing order. Hence, for the second-largest singular value $\sigma_2$ of $W_N$, we have

$$W_N V[2] = \sigma_2 U[2] \qquad \text{and} \qquad W_N^\mathsf{T} U[2] = \sigma_2 V[2] \tag{11}$$

where $U[2]$ and $V[2]$ denote the second columns of $U$ and $V$ respectively. Defining $f_u = U[2]$ and $f_v = V[2]$, we identify $f_u$ and $f_v$ as the left and right singular vectors associated with $\sigma_2$. The above relations can then be written compactly as

$$W_N f_v = \sigma_2 f_u \qquad \text{and} \qquad W_N^\mathsf{T} f_u = \sigma_2 f_v. \tag{12}$$

Given a new observation $x$ to classify, we can use the learned parameters $\sigma_2$, $f_v$, $D_2^{-1/2}$ and the threshold $th$ (discussed below) to determine the partition in which $x$ falls. We define the scaling $T : x \mapsto x_N$ by

$$x_N = \frac{x}{\sqrt{\sum_i x_i}} \tag{13}$$

$T$ is a scaling defined for a given point, that, when applied to the whole $W$ matrix, becomes $T(W) = D_1^{1/2} W$. Note that we do not need to define an additional mapping for $D_2^{1/2}$, since it is specified for each given feature and is therefore "learned" directly from $W$. We then compute the projection score

$$s(x) = \frac{f_v^\mathsf{T} x_N D_2^{-1/2}}{\sigma_2}, \tag{14}$$

and assign $x$ to one side of the split according to whether $s(x) > th$ or $s(x) \leq th$.

A linear classifier can be defined as $\text{clf} : \mathbb{R}^m \to [-1, 1]$ that associates to $x_n$ the predicted class $\hat{y}$ given by

$$\hat{y} = \text{sign}(\frac{f_v^\mathsf{T} x_N D_2^{-1/2}}{\sigma_2} - th) \tag{15}$$

where $th$ is a threshold parameter.

While the scaling $T$ is necessary for extracting clusters, as previously established (Dhillon, 2001; Zha et al., 2001), it also compresses data points near the set $\{x \in \mathbb{R}^m | x = T(x)\}$, as illustrated in two dimensions in Figure 1.

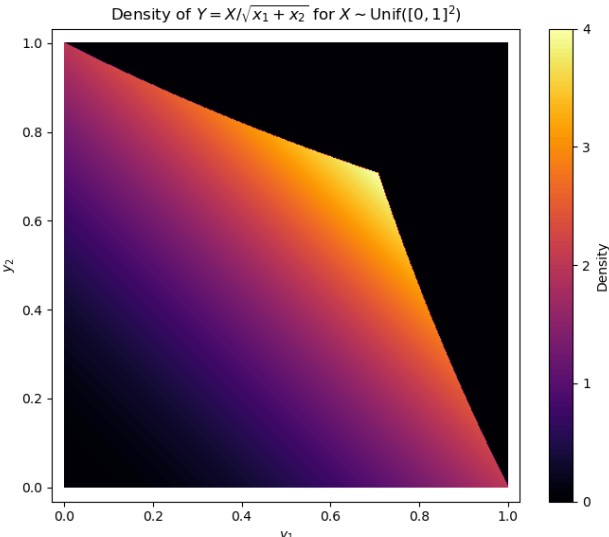

Figure 1: **Visualisation of the distortion induced by the function $T$ mapping $x$ to $x_N$.** Assuming $X \sim Unif([0,1]^2)$, and $Y = T(X)$, we plot $f_Y$, the density of $Y$.

### 2.3.2  SCUT hierarchical partitioning

$W$ is a non-negative weight matrix with $n$ observations and $m$ features. In this setting, the goal is to cluster the observations while retaining all $m$ features at each recursive step of the algorithm. To build the hierarchical partitioning, SCUT recursively splits the data by extracting the Fiedler vector $f_u$ from a matrix $W_N$, which is updated and recomputed at each step of the recursion with respect to the $m$ features using the SVD. Rather than splitting the data solely based on the sign of the components of $f_u$—a common practice that can lead to suboptimal or unstable clustering—SCUT introduces a flexible, data-driven strategy for determining a threshold $th$ that defines the partition.

Since the components of $f_u$ represent a linear projection of the $n$ observations (sampled from an $m$-dimensional space) onto a one-dimensional space, the vector $f_u$ can be interpreted as a sample from a one-dimensional distribution with probability density function $p$. In this context, regions of high density in $p$ correspond to modes of the projected distribution. Several such modes may coexist. SCUT does not attempt to isolate all modes simultaneously; instead, it identifies the most pronounced density valley and uses it to partition the data into two aggregated subsets, which are then recursively refined. Therefore, SCUT selects a threshold such that the split:

- occurs at a point of low density, and

- separates distinct high-density regions.

Locations that verify these conditions are usually called "valleys". The threshold $th$ is defined as:

$$th = \arg\max_z \left[ \left( \max_{x<z} \log \frac{p(x)}{p(z)} \right) \cdot \left( \max_{x>z} \log \frac{p(x)}{p(z)} \right) \right] \tag{16}$$

where, $\max_{x<z} p(x)$ and $\max_{x>z} p(x)$ correspond to the highest peaks of the density to the left and right of $z$, respectively. Intuitively, SCUT seeks a threshold located in a valley between two peaks, where the drop in log-density on both sides is significant. Note that valleys might not exist; in such cases, the above formula will yield a low density location. This is not necessarily undesirable, since valleys do not occur when attempting to split a homogeneous cluster. Let

$$h : z \to \left[ \max_{x<z} \log \frac{p(x)}{p(z)} \right] \cdot \left[ \max_{x>z} \log \frac{p(x)}{p(z)} \right] \tag{17}$$

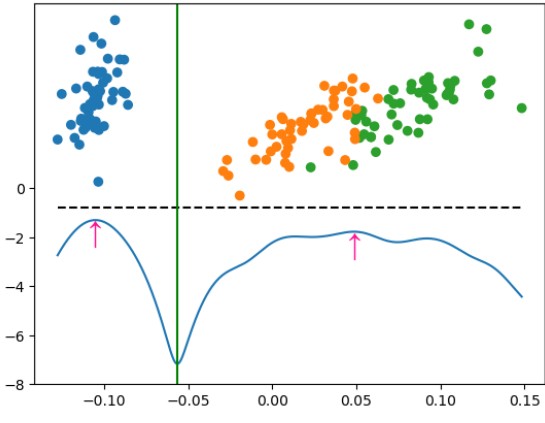

(a) First threshold selection on the Iris dataset

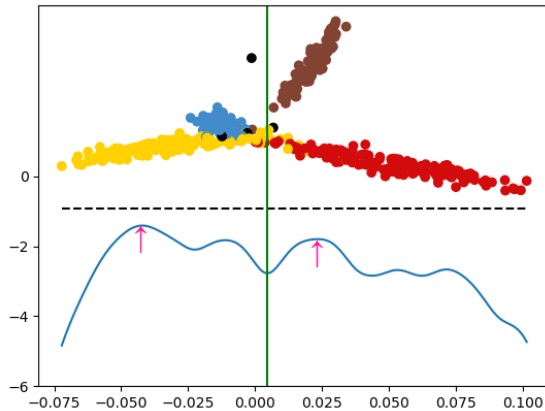

(b) First threshold selection on the French National Assembly dataset

Figure 2: **Illustrations of threshold selection.** (a): the Iris dataset. The $x$-axis represents the one-dimensional space containing $f_u = U[2]$. On the $y$-axis, below the dashed horizontal line: estimated log-density of $f_u$; above the dashed horizontal line: scatter plot of $(U[2], U[3])$, where $U[3]$ is used for visualization purposes only. Colors indicate species (blue: *I. setosa*, orange: *I. versicolor*, green *I. virginica*). The vertical green line marks the selected threshold $th$, and pink arrows highlight the maximum density peaks on the left and right of $th$. (b): the French National Assembly dataset. $x$-axis, $y$-axis, dashed horizontal line, vertical green line, and pink arrows as on left. Colors correspond to political alignment, "Non-attached" are in black.

where $p$ denotes the estimated density. We define a confidence metric

$$\mathrm{conf} = \max_{z} \left[ 1 - \frac{1}{e^{\frac{\sqrt{h(z)}}{2}}} \right] \tag{18}$$

which takes values in $[0, 1]$ and measures how well defined the clusters are. Intuitively, conf measures the quality of the split based on density: 0 indicates a poor split, and 1 indicates a perfect one. Note that the 2 in the denominator of the exponential serves as a scale parameter, controlling the slope of conf. Empirically, we found that using 2 worked well with our other hyperparameters. conf has the following properties:

- conf $= 1$ when $p(th) \to 0$ (i.e., a deep valley).

- conf $= 0$ when $p(th) = min(\max_{x<z} p(x), \max_{x>z} p(x))$ (i.e., no drop in density at threshold $th$)

- as a function of $p(z)$, in $[0, min(\max_{x<z} p(x), \max_{x>z} p(x))]$, conf is strictly increasing, being the composition of strictly monotone functions.

SCUT uses the log-density rather than the raw density because differences in log-density reflect relative changes—i.e., density ratios—which are more meaningful for identifying significant separations. Taking the product of the two differences emphasizes that both sides of the threshold should show strong contrast with the surrounding peaks; this models the logical "and" condition needed to detect a meaningful split.

The selection of the threshold is illustrated on Figure 2. On the Iris dataset, shown in Figure 2a, a threshold of 0 would misclassify some *I. versicolor* as *I. setosa*, whereas choosing a value at the density valley prevents such errors. In the French National Assembly dataset, shown in Figure 2b, multiple density valleys appear because the distribution contains several peaks—one for each political ideology, with multiple peaks within the left-aligned cluster. Selecting the threshold $th$ according to our criterion yields a split that correctly separates deputies from the majority and the opposition.

---

**Algorithm 1** HierarchicalSpectralClustering

---

**Require:** A positive data matrix $W$ and root node $t$
**Ensure:** A hierarchical clustering of $W$, starting at the node $t$.

1: **if** $W$ has only 1 row **then**
2:     $t.\text{is\_leaf} \leftarrow True$
3:     **return** $t$
4: **end if**
5: Compute $D_1$ and $D_2$ from $W$
6: $W_N \leftarrow D_1^{-1/2} W D_2^{-1/2}$
7: $U, \sigma, V \leftarrow \text{SVD}(W_N)$
8: $f_u \leftarrow U[2]$
9: Consider $f_u$ as a series of 1-d samples from a 1-d probability distribution :
10: $p \leftarrow$ a density estimation given $f_u$
11: $th \leftarrow \arg\max_z [[\max_{x<z} \log \frac{p(x)}{p(z)}] \cdot [\max_{x>z} \log \frac{p(x)}{p(z)}]]$
12: $l \leftarrow \{i \in U, f_u[i] < th\}$
13: $r \leftarrow \{i \in U, f_u[i] \geq th\}$
14: $t.left \leftarrow \text{HierarchicalSpectralClustering}(W[l], t.left)$
15: $t.right \leftarrow \text{HierarchicalSpectralClustering}(W[r], t.right)$
16: **return** $t$

---

Once the optimal threshold $th$ is selected, SCUT defines the two partitions:

$$X = \{u_i \in V \mid f_u[i] \geq th\} \quad \text{and} \quad Y = \{u_i \in V \mid f_u[i] < th\} \tag{19}$$

The same partitioning procedure is then recursively applied to the submatrices $W[\{i \mid u_i \in X\}]$ and $W[\{i \mid u_i \in Y\}]$ until each resulting matrix contains only a single observation. The complete recursive procedure is detailed in Algorithm 1.

### 2.3.3 SCUT inference for new observations

New data is frequently discovered, and researchers may wish to incorporate it into an existing hierarchical clustering. However, most hierarchical clustering methods do not support such updates and require the entire clustering to be recomputed—a process that is computationally expensive and may result in significant changes to the structure of the dendrogram. In contrast, during tree construction, SCUT stores at each internal node the linear decision function derived from the spectral split. This enables new observations to be routed through the existing hierarchy using standard tree inference, without recomputing the spectral decomposition or rebuilding the tree. Given a new data point $x$, the algorithm traverses the tree recursively: at each node $t$, it evaluates the stored decision function $clf_t(x)$ and assigns the point to the left or right child accordingly. If $clf_t(x) < 0$, the point is assigned to the left child $t.\texttt{left}$; otherwise, it proceeds to the right child $t.\texttt{right}$. This process is repeated until a leaf node is reached.

The complete inference procedure is presented in Algorithm 2. The traversal is structurally identical to standard decision tree inference; the difference lies in the fact that the branching rules are learned from spectral projections rather than from axis-aligned feature thresholds.

### 2.4 SCUT explainability

By construction, each node in the tree corresponds to a convex subspace of $\mathbb{R}^m$ that contains all data points associated with its descendant leaves. Note that this convexity holds in the space obtained after applying the scaling $T : x \rightarrow \frac{x}{\sqrt{\sum_i x_i}}$ (Figure 1). The root node covers the entire space $\mathbb{R}^m$. Let $e$ be an internal node, and let $S \subset \mathbb{R}^m$ denote the associated subspace. The left child $e.\texttt{left}$ corresponds to the region $\{x \in S \mid clf_e(x) < 0\}$, while the right child $e.\texttt{right}$ corresponds to $\{x \in S \mid clf_e(x) \geq 0\}$. These regions are convex by construction, meaning that for any $x, y \in S$, the line segment between $x$ and $y$ lies entirely within

---

**Algorithm 2** UnsupervisedDecisionTreeInference

---

**Require:** $x, t$,
**Ensure:** A predicted leaf for input $x$.
1: **if** $t.$is_leaf **then**
2:      **return** $t$
3: **end if**
4: $\hat{y} \Leftarrow clf_t(x)$
5: **if** $\hat{y} < 0$ **then**
6:      **return** UnsupervisedDecisionTreeInference$(x, t.left)$
7: **else**
8:      **return** UnsupervisedDecisionTreeInference$(x, t.right)$
9: **end if**

---

$S$. This convexity ensures that the clusters are geometrically coherent and stable under interpolation. Proof of this claim is detailed in the Supplementary Material.

Because each split corresponds to a linear decision function, the regions associated with internal nodes and leaves are convex in the transformed space. Consequently, highly non-convex cluster geometries may require multiple recursive splits to be approximated. However, the hierarchical composition of linear partitions produces a piecewise-linear decision boundary, analogous to classical decision trees, and is capable of approximating complex structures through recursive refinement. This design reflects a deliberate tradeoff: SCUT favors convex, geometrically coherent subspaces that enable interpretability, stability, and efficient inductive routing of new samples, rather than directly recovering globally non-convex clusters in a single step.

Further insight into the interpretability of SCUT can be gained from a physical analogy. Following the interpretation of spectral clustering as the analysis of a mass-spring system under transverse vibrations (Demmel, 1999; Pothen et al., 1990), the Fiedler vector $f_u$ can be seen as describing the displacement of the "observation nodes" at the second lowest resonating frequency during vibration. While $f_v$ describes the displacement of the "feature nodes." Since the positions of the feature nodes directly influence how strongly they "pull" on the observation nodes, this relationship offers an intuitive explanation of feature importance.

Given a new point $x$ to be explained in terms of the clustering decision, and using the normalized form $x_n = \frac{x}{\sqrt{\sum_i x_i}}$ as defined previously, the element-wise product $f_v \odot (x_n \, D_2^{-1/2})$ quantifies the influence—or "pull"—each feature exerts on $x$ during the partitioning step. This provides a mechanism for ante-hoc interpretability by revealing which features were most influential in assigning $x$ to a particular branch of the tree.

## 2.5 Metrics to validate the predictions

The SCUT algorithm constructs the dendrogram in a fully unsupervised manner and does not use ground-truth labels at any stage of training or tree construction. All evaluations against available labels are therefore performed strictly *post hoc*. To assess agreement with ground-truth labels, we employ two annotation procedures, Top-Down (TD) and Bottom-Up (BU), which map a fixed hierarchy to flat label predictions. These procedures operate solely on the learned dendrogram and do not modify its structure. The same TD and BU protocols are applied identically to all compared methods to ensure a fair evaluation.

The flat label assignments produced by TD and BU are evaluated using standard classification metrics: accuracy (ACC) and Matthews correlation coefficient (MCC). Let:

- $c$ be the total of correctly predicted samples,

- $s$ be the total number of samples,

- $t_i$ be the number of samples truly belonging to class $i$,

- $p_i$ be the number of samples predicted to belong to class $i$.

Accuracy is defined as

$$ACC = \frac{c}{s} \tag{20}$$

For the multiclass setting, we use the generalized Matthews correlation coefficient:

$$MCC = \frac{cs - \sum_i t_i p_i}{\sqrt{\left(s^2 - \sum_i p_i^2\right)\left(s^2 - \sum_i t_i^2\right)}}. \tag{21}$$

MCC takes values in $[-1, 1]$, where 1 indicates perfect prediction, 0 corresponds to random prediction, and $-1$ indicates total disagreement. Unlike accuracy, MCC remains informative under class imbalance and therefore provides a complementary assessment of predictive quality.

In addition to these flat prediction metrics, we report a hierarchy-level evaluation measure, dendrogram purity. This metric directly assesses structural alignment between the learned hierarchy and ground-truth labels without any label-dependent parameter tuning. It provides an evaluation of hierarchical coherence independently of the TD/BU mapping procedures.

### 2.6 Datasets

Experiments were conducted on five datasets spanning different data types: floral phenotype measurements from the Iris dataset (retrieved using scikit-learn's `sklearn.datasets.load_iris()` (Pedregosa et al., 2011)) (Fisher, 1936), textual data from the 20 Newsgroups dataset (retrieved using the scikit-learn library's `sklearn.datasets.fetch_20newsgroups()`) (Lang, 1995), political voting data from the French National Assembly[1], sequence data from the Glycoside Hydrolase protein family 30 (GH30) dataset[2] (Drula et al., 2022), and haplotypes from the *Human Genome Dataset (HGD) (1000 Genomes Project Consortium et al., 2015)*[3]. A sixth dataset is derived from user–item interactions on the Goodreads platform (Wan & McAuley, 2018)[4]. Unlike the other datasets, which serve as labeled clustering benchmarks, this example does not provide reliable ground-truth labels for users. Consequently, it is not used for quantitative evaluation but instead illustrates SCUT's ability to uncover latent structure in a large bipartite interaction matrix, revealing groups of books with similar readership patterns and clusters of users with similar reading preferences.

The structural characteristics of the six datasets are detailed in Table S1. For each dataset, we detail the preprocessing steps:

*Iris.* We first scaled the four original features to the $[0, 1]$ range using min–max normalization. To enrich the feature space, we generated four additional features by computing $1 - x$ for each normalized feature. The original and complemented features were then concatenated, yielding a dataset with eight features per observation, all in the $[0, 1]$ range.

*20 Newsgroups (20NG).* We applied TF-IDF vectorization to the text corpus, and removed terms (columns) whose total TF-IDF score across all documents (rows) was less than 2. This produced a sparse matrix of size $18,846 \times 11,812$—corresponding to $18,846$ documents and $11,812$ retained terms.

*French National Assembly (FNA).* We collected voting records from the 16th legislature of the French National Assembly, spanning June 22, 2022 to June 9, 2024. We retained all deputies who voted at least once, and all legislative votes, excluding senators who only participated in the constitutional vote on March 4, 2024. This yielded a dataset of 605 deputies and 4,106 votes. Since political alliances were prominent during this legislature, we assigned a political label to each deputy based on their party affiliation. Deputies were

---

[1] `https://data.assemblee-nationale.fr/archives-16e/votes`

[2] `https://www.lcqb.upmc.fr/profileview/documentation/gh30-usecase/`

[3] `https://gitlab.inria.fr/ml_genetics/public/artificial_genomes/-/tree/master?ref_type=heads`

[4] `https://cseweb.ucsd.edu/~jmcauley/datasets/goodreads.html`

categorized into five groups: left-wing, center, right-wing, far-right, or non-attached. Political nuances for each parliamentary group were derived from the official 2023 classification[5]. The data are represented as a $605 \times 4106$ matrix encoding each deputy's vote: 1 indicates a vote in favor, 0 indicates a vote against, and abstentions or absences are assigned the value $\frac{0.5+r}{2}$, where $r \in \{0,1\}$ denotes the outcome of the vote (0 if rejected, 1 if adopted). This encoding assigns missing votes a value close to the final outcome of the vote.

*GH30 dataset (GH30).* The GH30 dataset contains protein sequences belonging to Glycoside Hydrolase Family 30 (GH30), which is subdivided into several subfamilies according to their enzymatic function (Drula et al., 2022). We used vectorized representations of the 1803 sequences using the pipeline described in Vice-domini et al. (2022). After sequence and model filtering, the dataset contained 1675 sequences and 433 models (profile Hidden Markov Models). The pipeline produces two bit scores per model. We retained both scores and transformed them into values in $(0,1)$ using $S_i = \frac{1}{1+2^{-\frac{bs_i - median_i}{\sigma_i}}}$, where $i$ indexes the bitscore $bs_i$ ($2 \times 433$ scores in total), $median_i$ is the median of $bs_i$, and $\sigma_i$ is the standard deviation of $bs_i$. Because the GH30 dataset contains highly similar protein sequences, large bit scores are common. As a result, the transformation $S_i = \frac{1}{1+2^{-bs_i - meadian_i}}$ yields values that are effectively binary in practice (close to 0 when there is no match and close to 1 when there is a match). To obtain a more informative score distribution, we normalize the bit scores by their standard deviation, which produces a distribution with a median value of 0.5.

*Human Genome Dataset (HGD).* The HGD project comprises 5008 haplotypes from 2504 individuals, spanning 805 SNPs from all chromosomes. Only biallelic SNPs are included and compared to the reference human genome. For each haplotype, the alternative allele is tagged as 1 and the reference as 0. As in Iris, our preprocessing consist in concatenating the original feature vector to its complement.

*Goodreads.* The Goodreads dataset comprises user-book interaction records collected from the Goodreads platform. Metadata are available for books, including user-generated genre annotations obtained via a simple keyword-matching procedure, whereas no metadata are provided for users. Due to the high sparsity of the interaction matrix, we applied core filtering to retain approximately 28,000 users and 13,000 books.

## 3  Results

We evaluate SCUT across multiple datasets spanning textual, political, collaborative filtering, and genomic domains. Our objectives are to assess (i) its ability to recover meaningful hierarchical structure, (ii) its agreement with available ground-truth labels, (iii) its ability to inductively route new observations within the learned hierarchy, and (iv) its practical computational behavior. We compare SCUT to established hierarchical clustering methods, including classical agglomerative approaches and prior spectral divisive techniques. Across these settings, SCUT consistently matches or outperforms competing methods while maintaining competitive computational efficiency.

The divisive hierarchical clustering will be represented as a binary tree, where each leaf corresponds to an observation. To compare two such trees built on the same dataset, we developed two evaluation methods: a top-down annotation method (TD) and a bottom-up annotation method (BU). Both approaches assign a "predicted" class to each leaf; given a leaf, the two methods find a subtree that contains it and predict a class for it by returning the most common label in the subtree (excluding the leaf itself).

BU selects the smallest subtree containing the given leaf that includes at least $k$ leaves, where $k$ is a user-defined hyperparameter. The predicted label is the most frequent class among the other leaves in this subtree. The parameter $k$ is part of the evaluation protocol and determines the minimal subtree size used when mapping hierarchical structure to flat label predictions. It does not influence the construction of the dendrogram itself. The same value of $k$ is used across all compared methods to ensure a fair evaluation. Specifically, we set $k = 5$ for the Iris dataset, $k = 15$ for the French National Assembly dataset, $k = 20$ for the 20 newsgroups dataset, $k = 4$ for the GH30 protein family dataset, and $k = 20$ for the Human Genome Dataset.

---

[5]https://www.legifrance.gouv.fr/circulaire/id/45472

In contrast, TD starts at the root of the tree and traverses downwards. At each node, it compares the label distribution of its full subtree to that of its child subtrees. If the Kullback-Leibler (KL) divergence between the overall distribution and that of the child containing the target leaf exceeds a threshold $\theta$, the traversal continues to that child. This process is repeated recursively until the KL divergence falls below $\theta$, and the label distribution of the final subtree is used to assign a predicted class to the leaf. This procedure create a partition of the leaves into clusters, some cluster might contain only a few leaves and as such are probably outliers. We consider clusters having less than $k$ leaves to be outliers, using the same $k$ as in BU. For evaluation purposes only, for each fixed dendrogram, we selected $\theta$ to maximize ACC. Importantly, this selection is performed after the hierarchy has been fully constructed and does not influence the clustering itself. The same TD protocol and $\theta$-selection procedure are applied identically to all methods under comparison. Thus, the tuning affects only the external label assignment used for evaluation and not the learned hierarchical structure.

In addition to TD/BU-based accuracy metrics, we report dendrogram purity evaluation that does not involve any label-dependent parameter tuning. These metric assesses hierarchical alignment directly and confirms SCUT's structural advantage.

### 3.1 SCUT on the Iris dataset

The Iris dataset Fisher (1936) is one of the most widely used benchmarks in machine learning. It contains measurements of 150 Iris flowers, each belonging to one of the three species: *Iris setosa*, *Iris versicolor*, *Iris virginica*. Each flower is described by four features: petal length, petal width, sepal length, and sepal width. We applied min-max normalization to scale all features to the $[0, 1]$ range. To enrich the representation, we introduced four additional features by subtracting each normalized feature from 1. This transformation offers a complementary interpretation of each feature. For example, the original "petal length" feature yields two perspectives: "long petal" and "short petal".

SCUT produces a tree that perfectly clusters the *I. setosa* class (Figure S1A). While some mixing occurs between *I. versicolor* and *I. virginica*, SCUT largely recovers the underlying structure of the dataset. This mixing appears because the *I. versicolor* and *I. virginica* clusters almost overlap, with no density valley separating them. In contrast, the algorithm of Zha et al. (2001) splits not only observations but also features, and since there are only eight features, the resulting tree is influenced by too few of them. This limits expressiveness and leads to poor clustering (Figure S1B).

Intuitively, BU and TD scores reflect how well the clustering captures local and global structures, respectively. For BU and TD, we set the hyperparameter setting the minimal number of leaves in a subtree $k$ at 5 (see above).

On this toy dataset, both the quantitative metrics in Table 1 (top) and the dendrogram visualizations in Figure S1 demonstrate that SCUT more effectively captures both the global and local structure of the data compared to Zha et al. algorithm.

Note that in the TD approach, Iris clusters should have $\geq 5$ leaves and this leaves some leaves unlabelled in the tree of Figure S1.

### 3.2 SCUT on voting data from the French National Assembly

The voting behavior of French lawmakers offers a rich and nuanced dataset for clustering, as individual voting choices are shaped by both personal views and broader ideologies, party, and coalition alignments.

The 16[th] legislature of the French National Assembly exemplifies this complexity. During the recent elective term, there were 577 parliamentary seats. Of these, 250 were aligned with the governing majority, distributed among three political groups: Renaissance (RE, 169 seats), Democratic Movement (DEM, 50), and Horizons (HOR, 31). The opposition held 320 seats, the majority of which belonged to the NUPES coalition—including France Insoumise (LFI, 75), Socialist Party (SOC, 31), Communist Party (GDR, 22), and Europe Ecology – The Greens (ECO, 21). The remaining opposition seats were held by the National Rally (RN, 88), The Republicans (LR, 61), and the LIOT group (22). Note that although LR officially positions itself as an

| Method Annotation | SCUT | Zha Bottom-Up | Ward | SCUT | Zha Top-Down | Ward |
|---|---|---|---|---|---|---|
| THE IRIS DATASET | | | | | | |
| MCC | **0.93** | 0.72 | 0.92 | 0.93 | 0.72 | **0.94** |
| ACC | **0.95** | 0.81 | **0.95** | 0.95 (0.93) | 0.81 | **0.96 (0.94)** |
| THE FRENCH NATIONAL ASSEMBLY DATASET | | | | | | |
| MCC | **0.94** | 0.88 | **0.94** | **0.94** | 0.80 | **0.94** |
| ACC | **0.96** | 0.92 | **0.96** | **0.96** | 0.86 | **0.96** |
| THE 20NEWSGROUPS DATASET | | | | | | |
| MCC | **0.71** | 0.54 | 0.60 | **0.70** | 0.41 | 0.51 |
| ACC | **0.77** | 0.63 | 0.68 | **0.75 (0.72)** | 0.52 (0.45) | 0.60 (0.53) |
| THE GH30 PROTEIN FAMILY | | | | | | |
| MCC | 0.98 | 0.83 | **0.99** | **0.97** | 0.74 | 0.89 |
| ACC | 0.98 | 0.86 | **0.99** | **0.98 (0.97)** | 0.77 (0.76) | 0.91 (0.90) |
| THE HUMAN GENOME DATASET | | | | | | |
| MCC | **0.87** | 0.63 | 0.86 | **0.87** | 0.75 | **0.87** |
| ACC | **0.89** | 0.70 | **0.89** | **0.89** | 0.80 | **0.89 (0.86)** |

Table 1: **Comparative performance across five datasets of varying classification difficulty.** Classification performance of SCUT, the method of Zha et al. (2001) and Ward's method (Ward Jr, 1963) over the two annotation strategies TD and BU. The TD columns show scores without outliers and scores with outliers counted as misprediction in parentheses only if the 2 score are different. For the Iris dataset, the minimal cluster size is set to $k = 5$. For the French National Assembly dataset, $k = 15$, corresponding to the minimal number of lawmakers required to form a parliamentary group; outliers predicted by TD are assigned to the "non-attached" class. For the 20Newsgroups dataset, $k = 20$. For the GH30 dataset, $k = 4$ because the smallest class has only 4 labelled leaves. For the HGD, $k = 20$.

opposition party, its voting behavior frequently aligns with the majority. Seven seats were unassigned to any political group.

The dataset comprises 4,106 roll-call votes. For each vote, we record the position of every present lawmaker: in favor, against, or abstention. Due to substitutions and temporary replacements, the dataset includes 605 unique lawmakers, exceeding the number of official seats. Notably, in this legislature—characterized by the absence of a clear majority and the presence of fragile alliances—abstention or non-participation often served as a deliberate political signal. For example, a lawmaker might choose not to vote as a way of expressing disagreement with a proposed law without actively opposing its adoption. (See Methods for the encoding of abstentions or absences.)

For visualization and evaluation, each lawmaker is labeled according to the political leaning of their group: left-wing, center, right-wing, far-right.

Both clustering algorithms successfully recover the overall structure of the voting data. The initial split in both trees (Figure 3AB) clearly separates government-aligned groups from the opposition. While the LR group officially identifies as part of the opposition, many political analysts consider it generally aligned with the government—a nuance that is correctly reflected by both clustering methods. Within the opposition, both algorithms further distinguish between the far-right National Rally (RN) and the left-wing NUPES coalition. On the government side, both methods identify a distinct subcluster corresponding to LR, as well as a larger cluster representing the formal majority parties.

As shown by the BU and TD predictions scores (in Table 1, middle), SCUT effectively captures both the local and global structures in the voting data. Using the TD annotation method, SCUT is even able to correctly identify unaffiliated (non-attached) lawmakers based solely on their voting patterns. Moreover, SCUT's misclassifications tend to occur between politically adjacent groups-for example, many LIOT (centre) lawmakers are misclassified as left-leaning. Notably, only a single RN (far-right) lawmaker is incorrectly predicted to be a centrist. While the algorithm by Zha also captures much of the local and global structure,

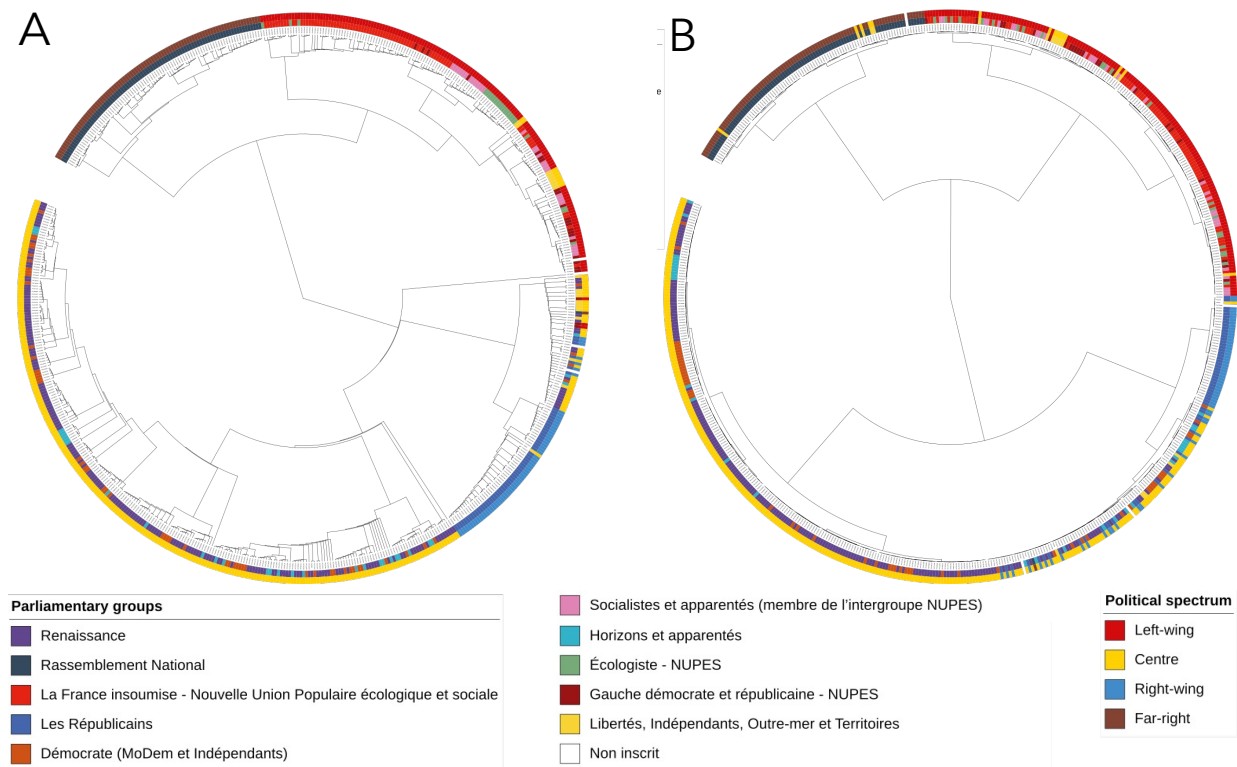

**Parliamentary groups**

- 🟪 Renaissance
- ⬛ Rassemblement National
- 🟥 La France insoumise - Nouvelle Union Populaire écologique et sociale
- 🟦 Les Républicains
- 🟧 Démocrate (MoDem et Indépendants)

- 🟪 Socialistes et apparentés (membre de l'intergroupe NUPES)
- 🟦 Horizons et apparentés
- 🟩 Écologiste - NUPES
- 🟥 Gauche démocrate et républicaine - NUPES
- 🟨 Libertés, Indépendants, Outre-mer et Territoires
- ⬜ Non inscrit

**Political spectrum**

- 🟥 Left-wing
- 🟨 Centre
- 🟦 Right-wing
- 🟫 Far-right

Figure 3: **Hierarchical clustering of the French National Assembly dataset.** Comparison of hierarchical clusterings produced by the SCUT algorithm (A) and the Zha et al. (2001) algorithm (B) on the French National Assembly dataset, comprising eleven Parliament parties/alliances organising four main political spectra. In both dendograms, the two concentric rings represent the ground-truth political alliances (inner ring) and the political leaning (outer ring) labels.

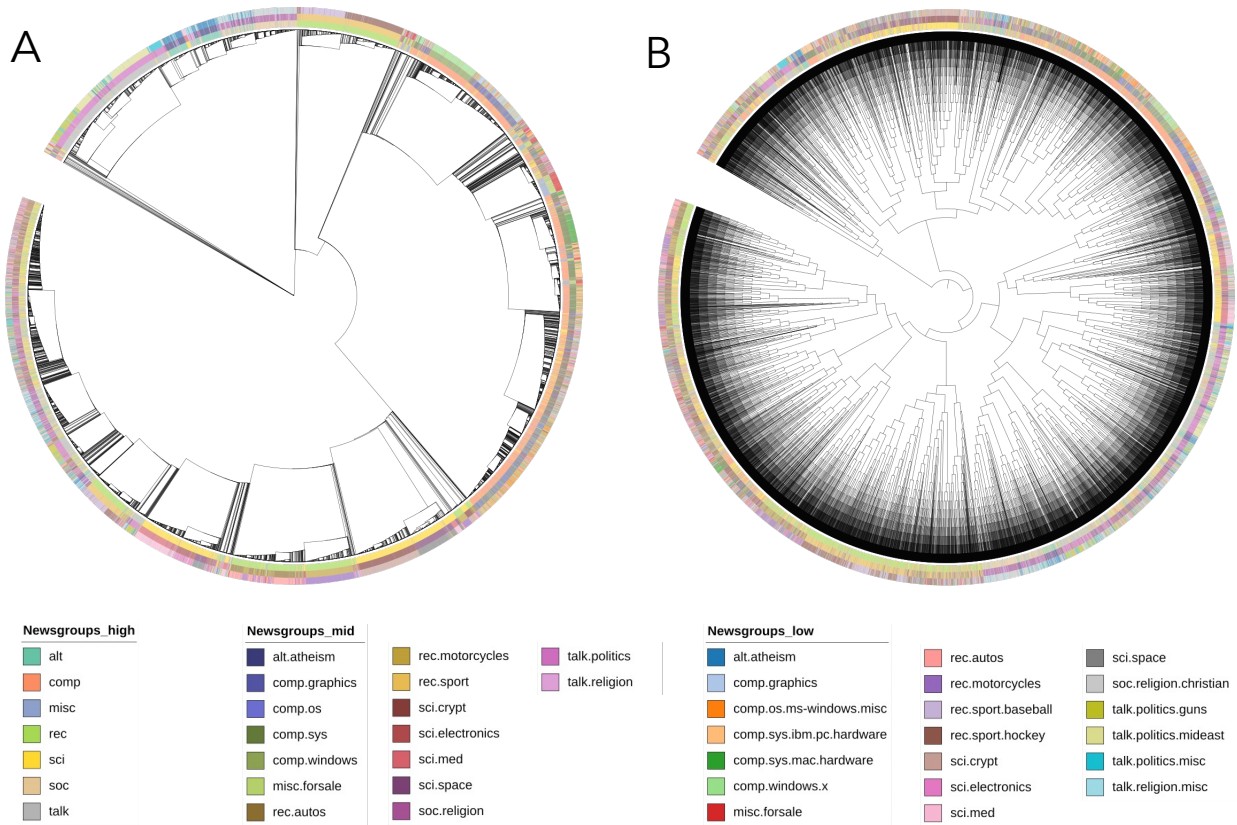

Figure 4: **Hierarchical clustering of the 20Newsgroups dataset.** Comparison of hierarchical clusterings produced by the SCUT algorithm (A) and the Zha et al. (2001) algorithm (B) on the 20Newsgroup dataset, comprising 7 discussion themes (inner ring), 16 broader themes (middle ring), 20 categories (outer ring). The discussion topics considered by the analysis are: computer-related topics (comp), scientific subjects (sci) recreational activities (e.g. games and hobbies; rec), socializing and social issues (soc), contentious issues such as religion and politics (talk), anything which does not fit in the other hierarchies (misc), alternative discussion (created after the ones above; alt).

its performance is consistently below that of SCUT. It exhibits a higher rate of misclassification, including more severe errors-for instance, in some cases, centrist lawmakers are predicted to belong to the far-right, and vice-versa. As illustrated in Figure 3, a significant portion of this discrepancy in performance stems from Zha's difficulty in correctly classifying LR (right-leaning) lawmakers, many of whom are incorrectly assigned to the centrist group.

### 3.3   20Newsgroup dataset

The 20newsgroups dataset is a collection of $18,846$ documents, each corresponding to a post from one of 20 Usenet newsgroups. These categories can be grouped into broader thematic areas; for example, the baseball and hockey newsgroups both fall under the general theme of sports.

To construct a weight matrix representing a bipartite graph between documents and terms, we applied TF-IDF vectorization to the text corpus. The resulting matrix was high-dimensional and sparse, with many terms appearing in only a few documents. To reduce noise and dimensionality, we removed columns (i.e., terms) whose total TF-IDF score across all documents (rows) was less than 2. This filtering step yielded a final matrix containing $11,812$ terms.

| Baseball | Score | Hockey | Score |
|----------|-------|--------|-------|
| duke | 0.0826 | gld | -0.1243 |
| econ | 0.0751 | cunixb | -0.0934 |
| bonds | 0.0747 | columbia | -0.0915 |
| fls | 0.0735 | dare | -0.0905 |
| batting | 0.0672 | gary | -0.0710 |
| adobe | 0.0637 | hrivnak | -0.0670 |
| alomar | 0.0636 | gtd597a | -0.0662 |
| mattingly | 0.0630 | buffalo | -0.0628 |
| hitter | 0.0620 | winnipeg | -0.0623 |
| williams | 0.0601 | espn | -0.0620 |

Table 2: Top 10 baseball and hockey terms with their relevance scores, positive scores indicate closeness to baseball, and negative ones to hockey.

For this dataset, predictions were made for the seven broadest Usenet categories, corresponding to the original seven top-level hierarchies (excluding `news.`) together with the `alt.` hierarchy (see legend in Figure 4AB). Zha's algorithm clearly struggles with this task: using the TD annotation method, it fails to assign any document to the `misc` or `soc` categories (Figure 4B). While SCUT also encounters difficulties in predicting `misc` documents, often misclassifying them as `comp`(Figure 4A), it nevertheless achieves significantly better performance overall, both in TD and BU scores (Table 1).

Examining the topology of the SCUT dendrogram in Figure 4A reveals several meaningful structures. A first large cluster, roughly spanning the 1 to 5 o'clock sector, is composed primarily of `comp` documents, although it also contains a few hundred `misc.forsale` documents. A second distinct `talk`/"religion" cluster is also clearly separated from the rest of the dendrogram (located in the 10 to 12 o'clock sector). This cluster further subdivides into `talk.politics` and "religion" subclusters. The `talk.politics` branch splits into `talk.politics.guns` and `talk.politics.mideast`, while the "religion" branch separates into `talk.religion.misc`, `alt.atheism`, and `soc.religion.christian`, albeit with some mixing. The topology of this `talk`/"religion" cluster illustrates SCUT's ability to capture both global and local organization within the dataset. However, this organisation is not reflected in the quantitative metrics because `talk.religion.misc`, `alt.atheism`, and `soc.religion.christian` correspond to distinct labels, even at the higher-level annotation scheme. SCUT also identifies a clear `rec.sport` cluster, which further separates into `rec.sport.baseball` and `rec.sport.hockey`, located between 12 and 1 o'clock. In the lower part of the dendrogram, roughly between the 5 and 7 o'clock sector, three additional clusters can be observed, corresponding predominantly to `sci.med` (left), `rec` (middle), and `sci` (right). The `rec` cluster further splits into `rec.autos` and `rec.motorcycles`, while the `sci` cluster divides into `sci.crypt` and `sci.space`. In contrast, in Figure 4B and Figure S2, both Zha's method and Ward's hierarchical clustering are able to group documents according to their high-level categories but fail to recover the finer-grained structure present in the low-level annotations, as reflected in Table 3.

We also examined the interpretability of the clusters produced by SCUT. For instance, in Figure 4A we consider the "sport" cluster located between the 12 and 1 o'clock sectors, which splits into two subclusters corresponding to hockey and baseball topics. We extracted the ten most influential terms for each of these subclusters (Table 2). In the "hockey" subcluster, several terms correspond to the email identifiers of highly active users (e.g., "gld", "cunixb"). Most of the remaining terms are strongly related to hockey, including team names such as "buffalo" (Buffalo Sabres) and "winnipeg" (Winnipeg Jets), as well as player names such as "hrivnak" (Jim Hrivnak). Other terms correspond to more general sports vocabulary, such as "espn", a sports network that covers both hockey and baseball. Interestingly, "espn" appears nearly ten times more frequently in the hockey newsgroup than in the baseball one. In the "baseball" subcluster, several terms again correspond to prominent users (e.g., "duke", "fls", "econ", "adobe"). Domain-specific terms include the names of players such as "mattingly" (Don Mattingly), "williams" (Billy Williams), "bonds" (Barry Bonds), and "alomar" (Sandy Alomar), as well as baseball-related vocabulary such as "batting" and "hitter". Overall, in both clusters the most influential terms align closely with the underlying topics, while a smaller subset

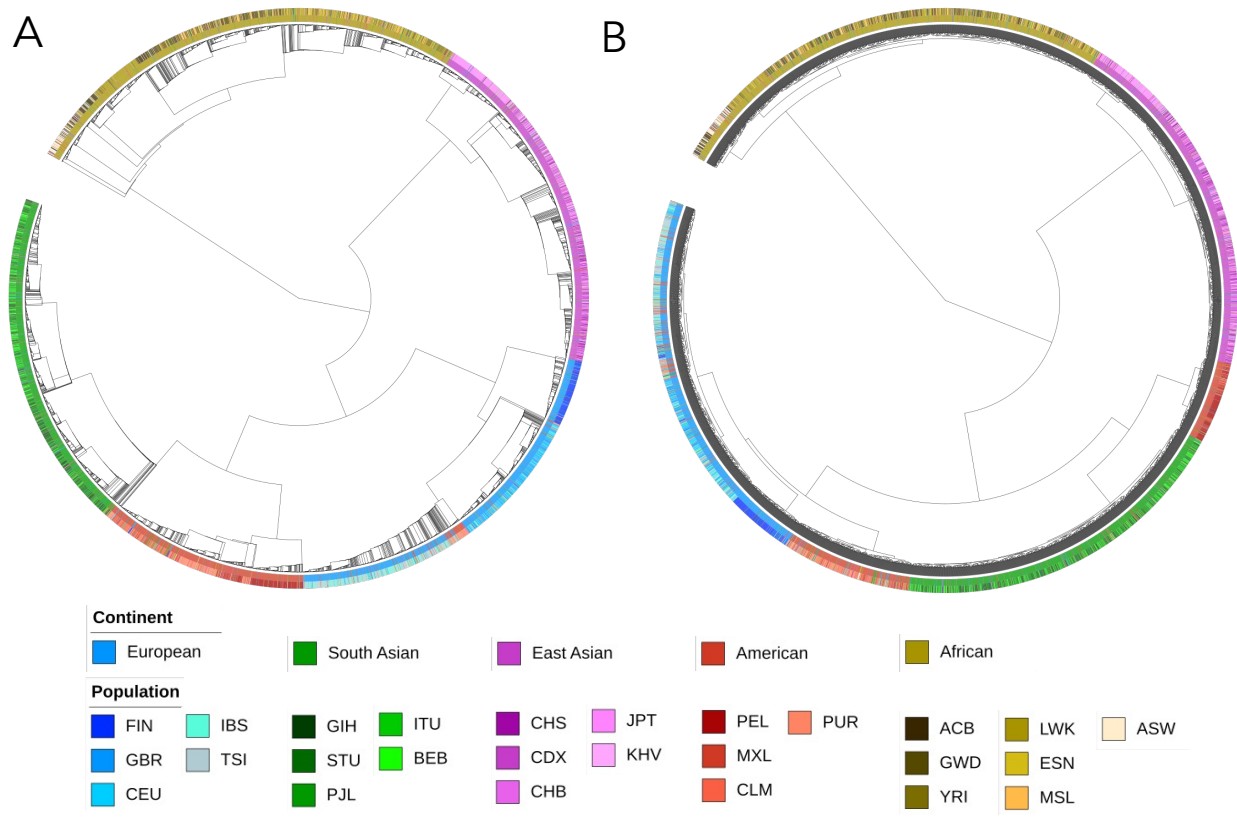

Figure 5: **Human Genome Dataset analysis with SCUT and Ward.** A. Dendrogram produced by SCUT. B. Dendrogram produced by Ward's agglomerative clustering. The inner ring represents continental origin (AFR, EAS, EUR, SAS, AMR), and the outer ring corresponds to individual population codes. Population labels described at: https://www.ensembl.org/info/genome/variation/species/populations.html

reflects highly active users in each group. These observations indicate that SCUT effectively captures the semantic structure of the dataset across different topics.

To assess the importance of these influential terms in structuring the clusters identified by SCUT, we performed a feature ablation experiment. Specifically, we removed all features whose weight magnitude exceeded 0.02 for the sport subcluster. This operation eliminated 555 features out of the 11,812 total terms. The topology of the resulting dendrogram (Figure S3) remains largely similar to the original one. However, two notable differences appear: the sport cluster is now positioned next to the other `rec` clusters, and, more importantly, the `rec.sport.hockey` and `rec.sport.baseball` documents are no longer clearly separated and instead become mixed. These results confirm that the influential terms identified above play a key role in distinguishing the two sport-related subclusters.

### 3.4 Analysis on the GH30 Dataset

The GH30 dataset consists of closely related protein sequences belonging to Glycoside Hydrolase Family 30 (GH30) Drula et al. (2022). This family is subdivided into nine subfamilies according to their specialized enzymatic functions. We used vectorized representations of 1,675 sequences, where each sequence is described by similarity scores against a collection of 433 profile Hidden Markov Models (HMMs), yielding two bit scores per model (Vicedomini et al., 2022). These scores were transformed into values in $(0, 1)$ using a sigmoid-based normalization in order to avoid the near-binary values produced by raw bit scores. The resulting matrix therefore encodes the similarity between sequences and functional motifs captured by the HMM profiles.

The annotation labels correspond to the nine CAZy GH30 subfamilies (Drula et al., 2022), which are manually curated. Most errors made by SCUT occur between sequences belonging to the GH30-4 and GH30-5 subfamilies (around 10 o'clock in Figure S4), and between GH30-1 and GH30-2 (around 5 o'clock). Interestingly, the dendrogram produced by SCUT achieves a dendrogram purity of 0.97, which is higher than the 0.95 reported for the GH30 dendrogram in (Vicedomini et al., 2022), obtained using Ward's clustering combined with Principal Component Analysis on the raw bit scores.

Using our preprocessing pipeline, Ward's method achieves the best accuracy and MCC on GH30 with the Bottom-Up annotation strategy, while retaining a dendrogram purity of 0.95. However, this strategy is biased because of the small value of $k$ ($k = 4$). Moreover, Ward's method fails to correctly recover the GH30-6 subfamily, which contains only four annotated sequences, splitting them into two separate groups. In contrast, SCUT groups these four sequences together, which is reflected in its higher dendrogram purity.

### 3.5   Analysis on the Human Genome Dataset

We evaluated SCUT on genomic data from the 1000 Genomes Project (1000 Genomes Project Consortium et al., 2015), a widely used benchmark for studying human genetic variation. The dataset comprises individuals sampled from 26 populations across five continental groups: Africa (AFR), East Asia (EAS), Europe (EUR), South Asia (SAS), and the Americas (AMR). We considered the Human Genome Dataset (HGD), consisting of 2,504 individuals (5,008 phased haplotypes) characterized by 805 biallelic single-nucleotide polymorphisms (SNPs) distributed across all chromosomes. These SNPs capture a substantial proportion of the population structure present in the full dataset. Each haplotype is encoded relative to the human reference genome as a binary vector of length 805, where 0 denotes the reference allele and 1 the alternative allele, yielding a binary matrix of dimension $5008 \times 805$.

To apply SCUT within our bipartite framework, each haplotype vector $\mathbf{v}$ was concatenated with its complement $\mathbf{1} - \mathbf{v}$, enabling the model to encode both similarity and dissimilarity to the reference genome. Individuals were labeled according to their continental origin for evaluation purposes.

SCUT recovers a globally coherent hierarchical structure that largely aligns with continental groupings (Figure 5A). Individuals from the same continent are predominantly clustered together, reflecting broad patterns of human genetic ancestry. Some overlap is observed between American and European populations, consistent with known historical admixture; for example, a subset of PUR (Puerto Rican) individuals is grouped within the European subtree. Within East Asia, the KHV (Kinh in Ho Chi Minh City, Vietnam) population forms a largely distinct subtree. European populations further organize into three principal substructures: one dominated by FIN (Finnish), a second grouping GBR (British) and CEU (Utah residents with Northern and Western European ancestry), and a third containing IBS (Iberian) and TSI (Toscani) populations. In the Americas, PEL (Peruvian) and MXL (Mexican Ancestry in Los Angeles) populations form coherent subtrees. African populations display the highest internal diversity, with LWK (Luhya in Webuye, Kenya) forming a distinguishable branch.

For comparison, Ward's agglomerative clustering applied to the original vectors $\mathbf{v}$ produces a broadly similar continental organization, but differs in its treatment of admixed American populations (Figure 5B). In particular, Ward separates American individuals into two subtrees, coupling them respectively with European and South Asian populations, whereas SCUT maintains a more coherent continental partition. When Ward is applied to the concatenated representation $(\mathbf{v}, \mathbf{1} - \mathbf{v})$, it is theoretically expected to yield an equivalent dendrogram under Euclidean distance, as the transformation preserves pairwise distances up to a constant factor. In practice, minor discrepancies may arise due to numerical effects. In our experiments, the concatenated representation—at the cost of approximately doubling computational time—produces a Ward dendrogram that more closely reflects the geographical structure, similarly to SCUT (Figure S5). Importantly, when continental origin is used as ground truth and the concatenated representation is adopted, SCUT achieves a dendrogram purity of 0.783 compared to 0.772 for Ward. When finer population labels are considered, SCUT attains a purity of 0.31, whereas Ward reaches 0.24 (see Table S2). These results indicate that SCUT more accurately captures both coarse-grained continental structure and finer-scale population organization. This improved structural coherence is consistent with the geometric properties of SCUT: the recursive linear

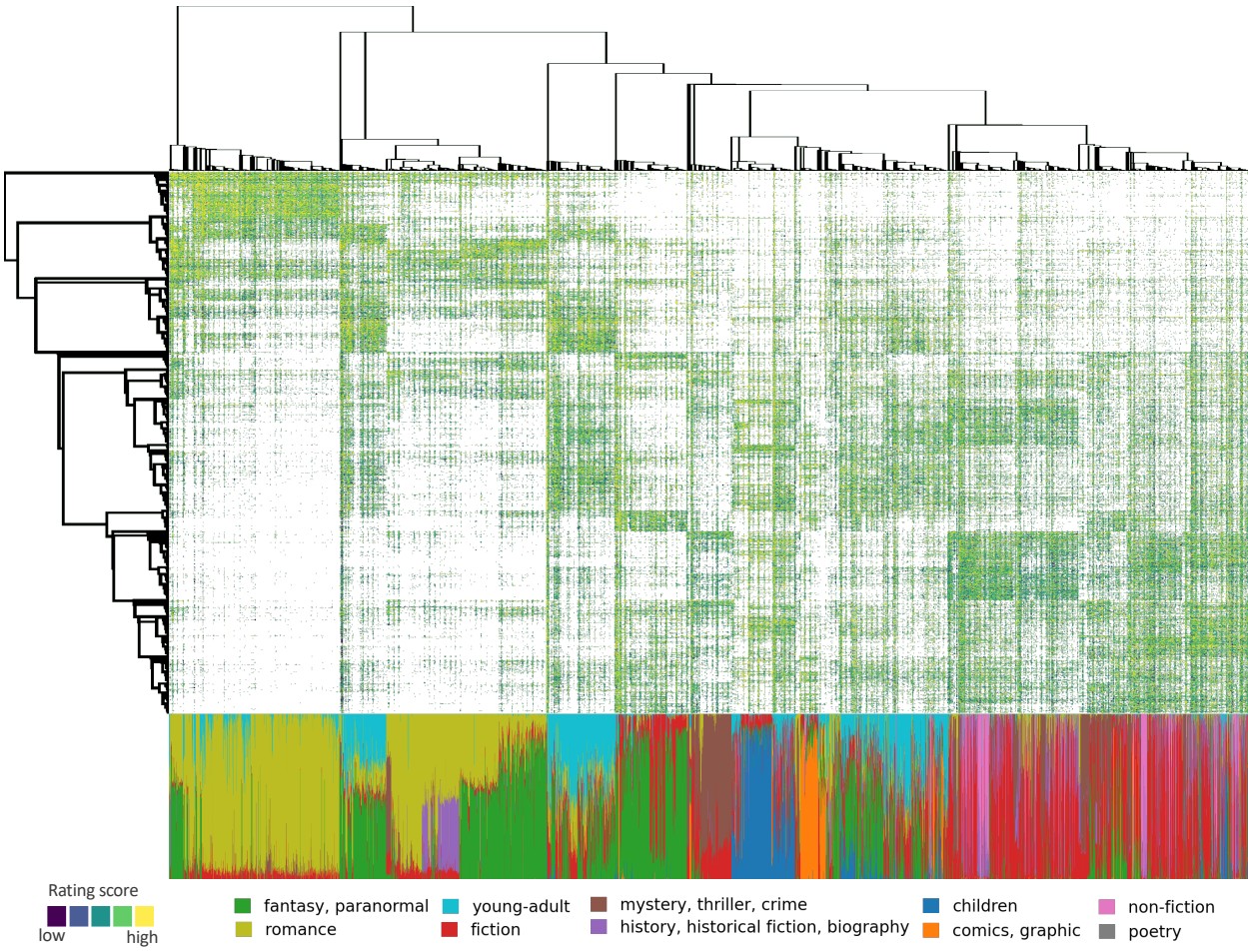

Figure 6: **Interaction patterns highlighted by SCUT.** The Goodreads interaction matrix (rows represent users, columns represent books) is ordered according to the SCUT trees. Each dot corresponds to a rating, colored using the viridis colormap (bright yellow = highest rating, dark blue = lowest rating). White cells indicate missing interactions. Books are labeled using user's reported tags: for example, if a book is tagged 20 times as 'children' and 10 times as 'fiction', its label bar is 2/3 blue (children) and 1/3 red (fiction). Tree branches are sorted by the number of leaves—at each node, the left child has fewer leaves than the right—to improve visual clarity. Because the original matrix ($13k \times 28k$) is too large to render directly, the figure uses a scatter-based representation of its nonzero entries. Note that overplotting may make the matrix appear denser than it truly is.

splits induce convex subspaces, promoting stable and well-separated partitions in high-dimensional binary genotype space.

## 3.6 Analysis on the Goodreads dataset

The Goodreads dataset Wan & McAuley (2018) is a large-scale collaborative filtering dataset comprising interactions between approximately 900,000 users and 2.4 million books. Each interaction corresponds to a user-assigned rating, taking integer values from 1 to 5, with 5 indicating the highest rating. In total, the dataset contains around 100 million such ratings.

Given the extreme sparsity of the full interaction matrix, we applied core filtering to obtain a denser submatrix comprising 9.7 million interactions between 28,000 users and 13,000 books. SCUT was then applied to

| Clustering approach | | Algorithm | 20NG high | 20NG low | FNA | GH30 | HGD cont | HGD pop | Iris |
|---|---|---|---|---|---|---|---|---|---|
| Graph based | Bipartite spectral | scut_kde | 0.50 | 0.30 | **0.93** | 0.97 | **0.78** | 0.31 | 0.82 |
| | | scut_kmeans++ | **0.52** | **0.38** | 0.90 | 0.89 | 0.76 | **0.33** | 0.83 |
| | | scut_median | 0.33 | 0.15 | 0.82 | 0.77 | 0.61 | 0.22 | 0.75 |
| | | scut_zero | 0.37 | 0.21 | 0.89 | 0.92 | 0.73 | 0.31 | 0.78 |
| | | Zha_clust | 0.34 | 0.16 | 0.88 | 0.89 | 0.67 | 0.17 | 0.73 |
| | Spectral | RecNCUTShiMalik | 0.43 | 0.29 | 0.89 | 0.89 | 0.70 | 0.19 | 0.68 |
| Distance based | Divisive hierarchical | bisecting k-means | 0.31 | 0.13 | 0.91 | 0.97 | 0.76 | 0.31 | 0.84 |
| | Agglomerative hierarchical | linkage_average | 0.23 | 0.07 | 0.77 | **0.98** | 0.75 | 0.22 | **0.85** |
| | | linkage_complete | 0.23 | 0.07 | 0.89 | 0.93 | 0.73 | 0.19 | 0.82 |
| | | linkage_ward | 0.33 | 0.16 | **0.93** | 0.95 | 0.76 | 0.24 | **0.85** |

Table 3: **Clustering performance comparison across clustering approaches.** Comparison of SCUT with baseline clustering methods from different families (divisive, spectral, agglomerative). Performance is reported using the dendrogram purity evaluation protocol on five datasets. For 20NG and HGD, results are reported for two label sets. bisecting k-means and scut_kmeans have been run with $k = 2$. Datasets: 20NG high-level (20NG high), 20NG low-level (20NG low), French National Assembly (FNA), GH30, HGD continental-level (HGD cont), HGD population-level (HGD pop), Iris.

this filtered matrix to construct hierarchical trees for both users and books. These trees were subsequently used to reorder the interaction matrix, thereby highlighting interaction patterns discovered by SCUT.

Book leaves in the resulting hierarchy were annotated using the genre metadata provided with the dataset Wan & McAuley (2018). These annotations were obtained through keyword matching in users' bookshelves (user-defined lists of books), as described in the original dataset documentation.

SCUT effectively extracts meaningful book genres using only the user-book interaction data. As illustrated in Figure 6, the book tree reveals well-defined clusters corresponding to distinct genre groupings. From left to right in the figure, the clusters include: a group dominated by romance titles; a smaller cluster combining fantasy/paranormal and young-adult books; a mixed cluster of romance works overlapping with other genres; a predominantly young-adult cluster with fewer fantasy/paranormal annotations than the preceding one; and a cluster composed mainly of fantasy/paranormal and fiction titles. This is followed by a smaller cluster rich in mystery/thriller/crime books with some fiction overlap. Next appears a large "young reader" cluster, which subdivides into three subgroups corresponding to children's books, comics, and a less clearly defined set. The remaining rightmost clusters encompass books spanning multiple genres—primarily fiction, non-fiction, and mystery/thriller/crime. Although this final cluster appears heterogeneous, the corresponding user tree reveals that the associated user groups exhibit broad yet internally consistent reading preferences. While external annotations for users were unavailable, qualitative inspection of the user clusters suggests that users within each cluster tend to share similar reading patterns and genre affinities.

### 3.7 Comparative evaluation across clustering approaches

To complement the dataset-specific analyses presented above, we compare SCUT with baseline clustering algorithms from several methodological families, including divisive spectral, classical spectral, divisive hierarchical, and agglomerative approaches. The comparison evaluates both the structural quality of the learned hierarchies (Table 3) and their computational cost (Table 4). Clustering quality is assessed using dendrogram purity, which measures how well the learned hierarchy aligns with ground-truth labels without relying on post-hoc label assignment.

Within the bipartite spectral family, we compare SCUT with the method of Zha et al. (2001), which performs recursive bipartite clustering using SVD but does not incorporate density-based threshold selection. As shown in Table 3, SCUT consistently achieves higher dendrogram purity across several datasets. This difference

| Clustering approach | | Algorithm | 20NG | FNA | GH30 | HGD | Iris |
|---|---|---|---|---|---|---|---|
| Graph based | Bipartite spectral | scut_kde | 288.16 | 3.39 | 4.98 | 34.52 | 0.32 |
| | | scut_kmeans++ | 253.58 | 5.41 | 12.25 | 41.09 | 1.31 |
| | | scut_median | 64.32 | 1.99 | 3.81 | 14.25 | 0.22 |
| | | scut_zero | 66.62 | 2.24 | 3.75 | 14.42 | 0.19 |
| | | Zha_clust | 34.85 | 1.27 | 1.09 | 2.99 | 0.07 |
| | Spectral | RecNCUTShiMalik | 230.73 | 19.03 | 9.37 | 27.21 | 0.64 |
| Distance based | Divisive hierarchical | bisecting k-means | 1180.46 | 8.25 | 14.37 | 39.19 | 1.14 |
| | Agglomerative hierarchical | linkage_average | 1554.52 | 0.48 | 0.71 | 7.18 | 0.00 |
| | | linkage_complete | 1581.49 | 0.39 | 0.73 | 7.32 | 0.00 |
| | | linkage_ward | 1533.49 | 0.38 | 0.81 | 6.42 | 0.00 |

Table 4: **Computational time comparison across clustering approaches.** Computational time (seconds) of the clustering methods evaluated in Table 3 (see legend), measured on the same datasets and experimental settings.

is particularly visible on datasets with heterogeneous or multimodal structure, suggesting that the density-aware split selection improves the quality of recursive spectral partitions.

Table 3 also reports several ablations of the thresholding rule within SCUT. In addition to the proposed KDE-based strategy (`scut_kde`), we evaluated three simpler alternatives under the same experimental protocol: `scut_zero`, which splits at zero following the standard sign-based partition used in spectral clustering; `scut_median`, which splits at the median of the Fiedler vector to enforce balanced partitions; and `scut_kmeans++`, where a one-dimensional $k$-means clustering ($k = 2$) with k-means++ initializationArthur & Vassilvitskii (2006) (best of 10 initializations), is applied to the Fiedler vector and the threshold is placed between the two centroids. These variants isolate the effect of the thresholding rule while keeping the rest of the SCUT pipeline unchanged. Across datasets, the density-aware strategy (`scut_kde`) generally achieves stronger or more stable dendrogram purity, particularly on structured datasets such as the French National Assembly and the Human Genome Dataset. Simpler thresholding rules tend to degrade performance when the projected distribution exhibits multiple modes or asymmetric density structure, indicating that part of SCUT's performance gain arises from the thresholding strategy rather than from the bipartite spectral split alone.

We also compare SCUT with classical spectral partitioning using the recursive normalized cut approach of Shi & Malik (2000). While this method captures global graph structure, it requires constructing and processing the full Laplacian matrix of size $(m+n) \times (m+n)$. In the bipartite setting considered here, this matrix contains two large zero blocks of size $(m \times m)$ and $(n \times n)$, which makes the approach computationally demanding for large datasets unless sparse representations are carefully exploited. In addition, the thresholding rule differs from SCUT: nodes are sorted according to their entry in the Fiedler vector, and the split is selected by maximizing the NCUT objective over all possible cut points.

Among divisive hierarchical baselines, bisecting $k$-means provides competitive results on several datasets, particularly when clusters are approximately spherical in the feature space. However, its reliance on Euclidean geometry limits its effectiveness in settings such as text or interaction data, where similarity is better captured by co-occurrence structure rather than geometric proximity.

Finally, we evaluate several agglomerative linkage methods (average, complete, and Ward). These approaches perform well on low-dimensional datasets with well-separated clusters, such as Iris or GH30. However, their performance deteriorates on high-dimensional sparse datasets such as 20 Newsgroups, where distance-based similarity becomes less informative. In these cases, methods exploiting the bipartite structure of the data tend to produce more coherent hierarchical organization.

The computational cost of the evaluated methods is reported in Table 4. As expected, agglomerative hierarchical clustering methods exhibit substantially higher runtimes on large datasets, reflecting their quadratic or cubic complexity in the number of observations. Zha's method remains the fastest among spectral approaches due to its simpler splitting rule. SCUT variants scale comparably to other divisive algorithms while

Table 5: Algorithm 2 stability on HGD (5-fold cross-validation)

| Method | ACC | MCC |
|---|---|---|
| SCUT + Algorithm 2 | **0.896 ± 0.007** | **0.868 ± 0.008** |
| k-NN | 0.818 ± 0.009 | 0.779 ± 0.010 |
| k-means + majority vote | 0.892 ± 0.026 | 0.864 ± 0.031 |

Table 6: Algorithm 2 performance on HGD (5-fold cross-validation)

| Labels | Dendrogram purity | Dendrogram purity with inserted samples | $|\Delta|$ |
|---|---|---|---|
| HGD Continents | 0.779 ± 0.005 | 0.778 ± 0.003 | 0.002 ± 0.001 |
| HGD Populations | 0.291 ± 0.008 | 0.283 ± 0.009 | 0.008 ± 0.002 |

maintaining strong clustering performance. Among the SCUT variants, the median and zero thresholding strategies are the fastest, while KDE-based thresholding incurs additional cost due to density estimation. Despite this additional cost, `scut_kde` remains substantially faster than agglomerative clustering on high-dimensional datasets such as 20 Newsgroups.

Beyond computational cost, `scut_kde` and `scut_kmeans++` consistently rank among the top-performing methods. In particular, `scut_kde` appears among the three best-performing approaches across nearly all datasets and annotation levels, with the exception of the Iris dataset. Although this dataset does not naturally correspond to a graph structure, both SCUT variants achieve performance comparable to strong distance-based clustering baselines, including bisecting k-means and hierarchical linkage methods.

The `scut_kmeans++` variant relies on a one-dimensional $k$-means clustering ($k = 2$) applied to the Fiedler vector to determine the split threshold. Because the result of $k$-means depends on initialization, we use the $k$-means++ initialization scheme Arthur & Vassilvitskii (2006) and retain the best solution over 10 runs. In the specific case of a one-dimensional partition with $k = 2$, the optimal solution could also be obtained exactly by sorting the samples and evaluating the Within-Cluster Sum of Squares (WCSS) for all possible cut points; however, this approach requires $\Theta(n \log n)$ time due to the sorting step.

Overall, these results indicate that SCUT provides a favorable trade-off between clustering quality and computational efficiency across datasets with diverse structural properties.

### 3.8   Out-of-sample insertion analysis

To evaluate the inductive routing capability of SCUT, we conducted an out-of-sample insertion experiment on the Human Genome Dataset. Algorithm 2's performance was computed using continent- and population-level annotations. The dataset was split using 5-fold cross validation. For each fold, the SCUT hierarchy was constructed using only the training data. Test samples were then inserted using Algorithm 2, which routes samples through the stored linear decision functions without recomputing the spectral decomposition. We used the BU annotation strategy, based on ancestor nodes with at least 10 leaves.

We evaluated three aspects of the insertion procedure:

(i) label agreement with ground truth annotations was measured using ACC and MCC (Table 5).

(ii) we compared the leaf assignment before and after insertion with the assignment obtained by rebuilding SCUT tree on the full dataset (Table 6 and Table 3). We used both the continent-level and population-level, annotations, and evaluated the dendrogram purity, on the trees built on the training set, on the trees after inserting the new samples, we measured the absolute change ($|\Delta|$) in dendrogram purity between the two trees. See Table 3 for the dendrogram purity of the SCUT tree built on all sequences.

(iii) we compared the runtime of inserting samples using Algorithm 2 with the runtime required to rebuild the clustering.

Table 6 shows that SCUT insertion achieves strong agreement with ground-truth labels while maintaining high consistency with the clustering obtained by rebuilding the tree (Table 3). Furthermore, insertion is several orders of magnitude faster than rebuilding, demonstrating the practical advantage of SCUT's inductive routing mechanism. The insertion of a thousand datapoints took $0.85s \pm 0.07s$. The SCUT out-of-sample insertion was compared to three baseline algorithms: k-means++ (with k=10), where new samples were inserted by choosing the closest cluster and where the predicted class is defined as the majority class in each cluster; k-nearest neighbors algorithm (with k=10).

### 3.9 Complexity analysis

Let the input matrix $W \in \mathbb{R}^{n \times m}$. First, note that computing $W_N = D_1^{-1/2} W D_2^{-1/2}$ can be done efficiently in $\mathcal{O}(mn)$ operations. To obtain a fast approximation of the truncated SVD of $W_N$, one may use the method proposed in Halko et al. (2011), which achieves a complexity of $\mathcal{O}(mnk + k^2(n + m))$ flops, where $k$ is the number of singular values retained. For small $k$, this is commonly written as $\mathcal{O}(mnk)$. In our case, we set $k = 2$—since only the Fiedler vector is required— reducing the complexity to $\mathcal{O}(mn)$. Alternatively, the IRLBA method Baglama & Reichel (2005) provides another efficient approach, running in $\mathcal{O}(mnk)$ flops.

The threshold $th$ is selected by evaluating the density function $p$ at $s$ evenly spaced points between the minimum and maximum of the Fiedler vector $f_u$. This step requires $\Theta(ns)$ operations using naive kernel density estimation, and $\Theta(n + s \log s)$ using Fast Fourier Transform (Silverman, 1982). We treat $s$ as a constant in the rest of the analysis. A more sophisticated approach to finding $th$, for example using algorithms from black-box optimization, could further reduce the overall computational time. In SCUT, in the worst case, each recursive step isolates a single data point, resulting in a time complexity of $\mathcal{O}(mn^2)$. In the best case, each recursive step divides the dataset into two balanced subclusters, leading to a complexity of $\Omega(mn \log n)$. This best-case and worst-case behavior is reminiscent of the Quicksort algorithm. However, unlike Quicksort, we cannot analytically derive an average-case complexity for SCUT, as the size of each recursive split depends on the data distribution and is not independent of the input structure.

The SCUT method can be modified to ensure a time complexity of $\Theta(mn \log n)$ in both the best and worst cases. This is achieved by constraining the size of the splits during the recursive partitioning process. Specifically, during the threshold selection step (line 10 in Algorithm 1), if the smaller of the two resulting clusters contain less than $\frac{n}{c}$ elements-where $2 \le c < n$ is a hyperparameter-the threshold $th$ is reselected such that the smaller cluster contains exactly $\lceil \frac{n}{c} \rceil$ elements. This reselection can be efficiently performed in linear time using the Introselect algorithm (Musser, 1997). With this modification, the worst case behavior is improved: at each recursive step, the algorithm guarantees that at least a $1/c$-fraction of the data is separated, ensuring that the recursion depth remains logarithmic. Consequently, the overall time complexity becomes $\mathcal{O}(mn \log n)$ in both the best and worst cases.

We now analyze the memory complexity of SCUT. Since each split is processed independently, the memory required by SVD computations is dominated by the decomposition performed on the original matrix. When computing the two largest singular vectors, truncated SVD algorithms require $\Theta(m + n)$ auxiliary memory (Halko et al., 2011). Algorithm 1 must also store the original data matrix $W$, which requires $\Theta(mn)$ memory. This matrix is passed by reference to recursive calls: each call extracts the subset of rows corresponding to the current cluster (this takes at most $\mathcal{O}(mn)$ space), performs the SVD on the resulting submatrix, and discards it after the split is completed. The threshold selection step based on Kernel Density Estimation requires $\Theta(ns)$ memory with a naive implementation and $\Theta(s)$ memory when using the FFT-based method. Since $s$ is treated as a constant, this contribution remains negligible compared to storing $W$. Finally, the resulting dendrogram contains $n - 1$ internal nodes (one per split). If we optionally store the linear decision boundary at each node as a vector in $\mathbb{R}^m$ together with the threshold $th$, storing the tree requires $\Theta(mn)$ memory. Overall, the space complexity of building the SCUT dendrogram is therefore $\Theta(mn)$.

## 4 Limitations and possible improvements

Although our Python implementation already includes several optimizations (see Appendix A.4), further improvements are possible. In particular, the algorithm could benefit from parallelization, either at the

level of the SVD computation—for instance by using GPU implementations of randomized SVD Halko et al. (2011)—or at the level of the recursive tree construction. In Algorithm 1, the recursive calls at lines 14 and 15 are independent and could therefore be executed in parallel.

Qualitatively, the main limitation of SCUT arises from the use of linear cuts, which restrict it to concave clusters. SCUT relies on linear (affine) splits at each node. While recursive composition allows piecewise-linear approximation of complex and non-convex cluster structures, highly curved or intertwined geometries may require deeper trees to be represented accurately. In contrast to flat spectral clustering methods that can recover non-convex clusters directly, SCUT prioritizes hierarchical organization, convex subspaces (in the transformed space), and native out-of-sample assignment. This design choice therefore trades global nonlinear partitioning for interpretability, scalability, and consistent inference.

One possible direction to address this limitation is to incorporate nonlinear embeddings before the recursive partitioning step. For example, approaches inspired by the kernel trick used in Support Vector Machines could allow the algorithm to operate in a transformed feature space where nonlinear cluster boundaries become linearly separable. Similarly, techniques such as Landmark-based Spectral Clustering Chen & Cai (2011) could provide scalable nonlinear representations that alleviate the limitations of linear splits while preserving the hierarchical structure of SCUT.

## 5 Conclusion

We introduced SCUT, a novel hierarchical clustering approach based on spectral clustering. We demonstrated its effectiveness across multiple and diverse datasets, showing that SCUT outperforms existing hierarchical clustering methods in capturing both global and local data structures. On textual data, we further illustrated its ante-hoc interpretability by analyzing cluster splits, showing that the most influential features align closely with the semantic structure of the data. Moreover, the explanations provided by SCUT inherit stability properties from the spectral cuts on which they are based. Each split is determined by the Fiedler vector of the graph Laplacian, which is unique (up to sign) when the second-smallest eigenvalue is simple, ensuring that the corresponding partition is uniquely defined. In practice, only minor numerical variations may arise from the eigensolver used to approximate this vector, rather than from instability in the SCUT methodology itself. Using an additional multi-label dataset, we also showed that SCUT can recover meaningful cluster organization in more complex settings.

In addition, we showed that SCUT achieves a lower theoretical best-case time complexity than widely used agglomerative clustering methods while maintaining the same worst-case guarantees. We additionally proposed minor modifications that reduce the worst-case complexity from quadratic to quasi-linear, strengthening its scalability for large datasets..

Beyond standard benchmarks, SCUT is particularly suited to settings where the primary challenge is the structured organization of large and continuously evolving databases. In domains such as genomics, proteomics, and other large-scale scientific repositories, clustering serves not only as an exploratory tool but as a mechanism for organizing knowledge. By combining hierarchical partitioning with principled out-of-sample routing, SCUT can function as a stable organizational layer that integrates new data without requiring global recomputation. Similar requirements arise in large document collections, recommendation systems, and high-dimensional representation spaces, where scalability, interpretability, and consistent sample assignment are essential for long-term maintenance and analysis.

## 6 Data availability

SCUT code and the data to reproduce the analysis described in this article are available at: `https://anonymous.4open.science/r/SCUT_review-EF00/`.

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

# A   Appendix

## A.1   Proof of the convexity of SCUT clusters

First, let's show that $C = \{x \in \mathbb{R}^m | f(x) < 0\}$ where $f$ is an affine function, is a convex subspace of $\mathbb{R}^m$.

*Proof.* The function $f$ is affine so we can write for any $x \in \mathbb{R}^m$, $f(x) = w^\mathsf{T} x + b$, where $w \in \mathbb{R}^m$ is the weight vector and $b \in \mathbb{R}$ the bias.

Let $x$ and $y$ two points in $C$. For any $\lambda \in [0, 1]$, consider $z = \lambda x + (1 - \lambda)y$.

$$
\begin{aligned}
f(z) &= w^\mathsf{T} z + b \\
&= w^\mathsf{T}(\lambda x + (1 - \lambda)y) + b \\
&= \lambda(w^\mathsf{T} x + b) + (1 - \lambda)(w^\mathsf{T} y + b) \\
&= \lambda f(x) + (1 - \lambda)f(y)
\end{aligned}
$$

Since $f(x) < 0$ and $f(y) < 0$, $f(z) < 0$. Hence $z \in C$, which proves $C$ is convex.   □

It can be similarly proven that $\{x \in \mathbb{R}^m | f(x) \geq 0\}$ is also convex.

Let's prove the convexity of all SCUT clusters. Given a non-leaf cluster $t$, we note its associated space $S$ and $S.left$ ($S.right$) the space associated to $t.left$ ($t.right$).

We prove by induction the convexity of $S$ for any $t$ SCUT cluster.

*Proof.* **Basis step :** $t$ is the root note, $S = \mathbb{R}^m$. By definition, $S$ is convex.
**Inductive step :** Let $t$ a non-leaf cluster, we note $clf$ the affine function that splits $t$, assume $S$ is convex.

$$
\begin{aligned}
S.left &= S \cap \{x \in \mathbb{R}^m | clf(x) < 0\} \\
S.right &= S \cap \{x \in \mathbb{R}^m | clf(x) \geq 0\}
\end{aligned}
$$

$S.left$ and $S.right$ are convex as the intersection of 2 convex sets.   □

Note that the clusters are convex on the sets containing the observations after applying the scaling $T : \mathbb{R}^m \to \mathbb{R}^m$, which associate $x$ to $\frac{x}{\sqrt{\sum_i x_i}}$.

## A.2   Density analysis of French National Assembly clusters

Using the same method used to build Figure 2, we can analyze quantitatively the quality of the SCUT clusters built on the French National Assembly dataset. Figure 7 shows the first few "majority" clusters. The first split (Figure 7a) is quite hard to make as there are many deputies, from all part of the political spectrum, that did not vote much and ends up close to 0 in the $f_u$ projection, anyway, SCUT still manages to make coherent clusters that split cleanly the data between "majority" and "opposition". Interestingly, while $U[2]$ encodes for "majority/opposition", $U[3]$ seems to encode for the "left/right" political spectrum. The next split (Figure 7b) singles out a single deputy that while non-attached was supported by the far-right. The last 2 splits (Figure 7cd) try to separate the "presidential majority" from "The Republicans".

The same analysis can be done on the opposition cluster (Figure 8) where the 2 clusters "NUPES" and "National Rally" are easily distinguishable.

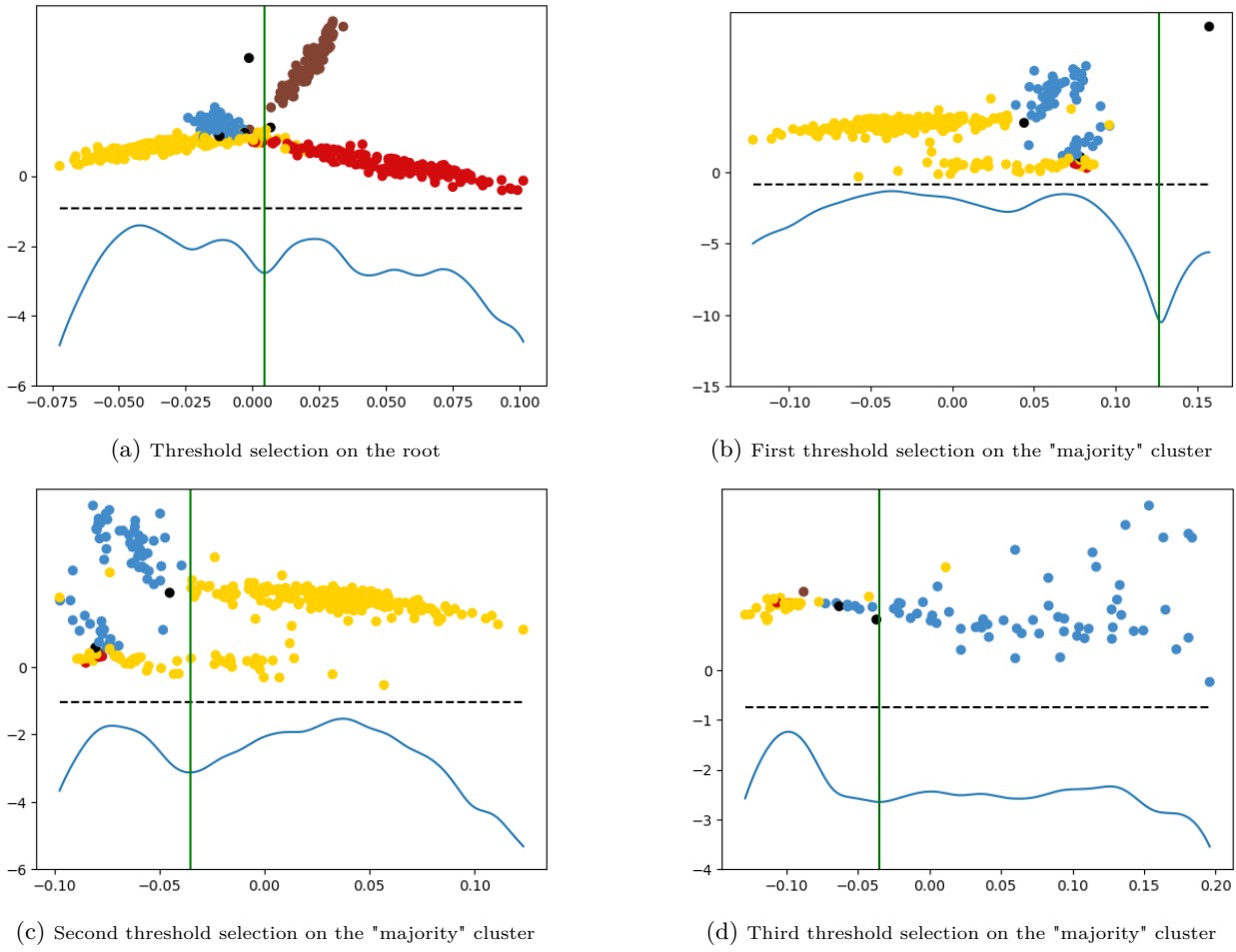

(a) Threshold selection on the root

(b) First threshold selection on the "majority" cluster

(c) Second threshold selection on the "majority" cluster

(d) Third threshold selection on the "majority" cluster

Figure 7: Threshold selection on the "majority" cluters of the French National Assembly dataset: $x$-axis : one-dimensional space containing $f_U = U[2]$. $y$-axis below dashed line : estimated log-density. above dashed line : scatter plot of $(U[2], U[3])$, $U[3]$ is used for visualization purposes only, colors correspond political alignment, "Non-attached" are in black. Vertical green line corresponds to the selected threshold.

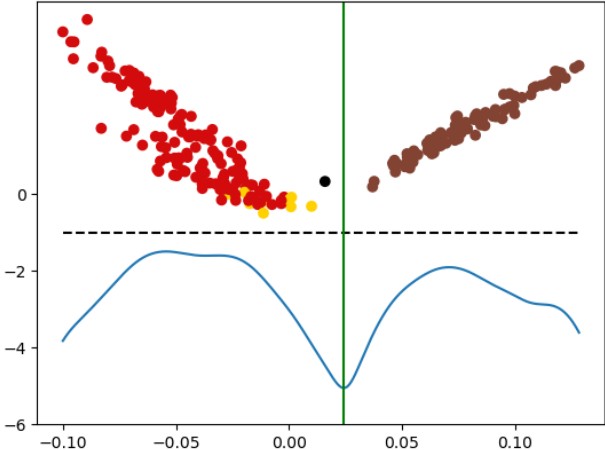

Figure 8: Threshold selection on the "opposition" cluster of the French National Assembly dataset : $x$-axis : one-dimensional space containing $f_U = U[2]$. $y$-axis below dashed line : estimated log-density. above dashed line : scatter plot of $(U[2], U[3])$, $U[3]$ is used for visualization purposes only, colors correspond political alignment, "Non-attached" are in black. Vertical green line corresponds to the selected threshold.

### A.3 Analysis of the bandwidth parameter

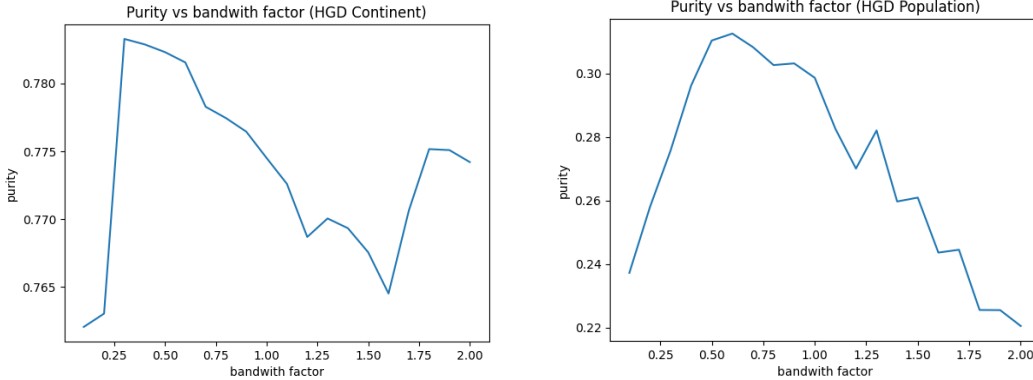

Figure 9: **Purity vs Bandwidth factor for continent- and population-level analyses of HGD.** Purity of SCUT trees built with different bandwidth factors is evaluated using HGD continent-level (left) and population-level (right) labels. SCUT default bandwidth factor is 0.5.

The threshold selection procedure relies on estimating the density of the Fiedler vector $f_u$ using kernel density estimation (KDE). Since KDE methods depend on a bandwidth parameter, we evaluated the sensitivity of SCUT to this hyperparameter. For a Gaussian kernel and a univariate Gaussian distribution, the bandwidth that minimizes the mean integrated squared error is given by Silverman's rule-of-thumb $h = 1.06\hat{\sigma}n^{-1/5}$ (Silverman, 2018), where $\hat{\sigma}$ is the empirical standard deviation of the samples. However, in our setting the projected distribution of $f_u$ is typically bimodal, reflecting the two clusters produced by a spectral split. To account for this, we define the effective bandwidth as $bw = bf \times h$, where $bf$ is a bandwidth factor controlling the amount of smoothing. To evaluate the robustness of SCUT with respect to this parameter, we varied $bf$ over a range of values and measured the dendrogram purity of the resulting hierarchies on the Human Genome Dataset using both continent-level and population-level labels. The results are shown in Figure 9. Across both evaluation settings, performance remains stable over a broad interval of bandwidth factors, with a shallow optimum around $bf \approx 0.5$. This value corresponds to the default parameter used in

SCUT. These results indicate that the KDE-based threshold selection is not highly sensitive to the precise bandwidth choice, and that the default value provides a good trade-off between smoothing and resolution.

### A.4   Practical implementation details

The SCUT implementation has been optimized in several ways to improve both runtime and memory efficiency

i. Handling of sparse matrices : Many data matrices contain many zeros, which are irrelevant for SVD computation, solvers like ARPACK(Lehoucq et al., 1998) allows for their efficient processing.

ii. Cherry-case optimization: small subproblems arising near the end of the recursion (e.g., clusters of size two) are handled explicitly, avoiding unnecessary SVD computations.

iii. Reduced memory usage: instead of storing multiple copies of the data matrix during recursion, effectively doubling memory consumption, the implementation maintains a single global data matrix and creates temporary local matrices only when required for SVD computations; these are released immediately after use.

iv. FFT-based KDESilverman (1982): threshold selection relies on density estimation of the one-dimensional Fiedler vector evaluated on equally spaced points. To accelerate this step, the kernel density estimate is computed using FFT-based convolution.

#### A.4.1   KDE implementation

Since the density is one-dimensional and evaluated on equally spaced points, we first build a histogram with $s$ bins over the $n$ samples, which requires $\mathcal{O}(n+s)$ operations. The histogram is then convolved with a Gaussian kernel of size $\mathcal{O}(s)$. This convolution can be efficiently computed using the Fast Fourier Transform, resulting in a complexity of $\mathcal{O}(s \log s)$. Consequently, the overall cost of the one-dimensional density estimation step is $\mathcal{O}(n + s \log s)$. For fixed $s$, this matches the asymptotic complexity of naive KDE, while providing a substantial practical speedup.

#### A.4.2   SVD algorithm

The singular value decomposition is computed using the ARPACK solver (Lehoucq et al., 1998). In our experiments, ARPACK proved faster on sparse matrices than the randomized SVD implementation available in scikit-learn (Pedregosa et al., 2011), which is based on the method of Halko et al. (2011).

## B   Appendix

### B.1   Supplementary Tables

| Dataset | Data type | Structure tested |
| --- | --- | --- |
| **Iris** | Low-dimensional continuous | Simple geometric clusters |
| **French Assembly** | Binary voting matrix | Political alignment structure |
| **20 Newsgroups** | High-dimensional sparse text | Semantic topic structure |
| **HGD** | High-dimensional binary genetics | Hierarchical population structure |
| **GH30** | Protein sequence similarity | Fine-grained functional clusters |
| **Goodreads** | Interaction matrix | User-item co-occurrence |

Table S1: **Datasets structural characteristics.**

| Method | Representation | Continental structure | Population structure | Time (in sec) |
|--------|:--------------:|:---------------------:|:-------------------:|:-------------:|
| **Ward** | **v** | 0.756 | 0.242 | 7.67s |
| | $(\mathbf{v}, \mathbf{1} - \mathbf{v})$ | 0.772 | 0.243 | 15.76s |
| **SCUT** | **v** | **0.788** | 0.262 | 31.94 |
| | $(\mathbf{v}, \mathbf{1} - \mathbf{v})$ | 0.782 | **0.310** | 32.33s |

Table S2: **Comparative purity scores across SCUT and Ward on HGD.** Purity scores for the SCUT method and Ward method (Ward Jr, 1963) over the HGD for coarse-grained continental structure and finer-scale population organization.

## B.2 Supplementary Figures

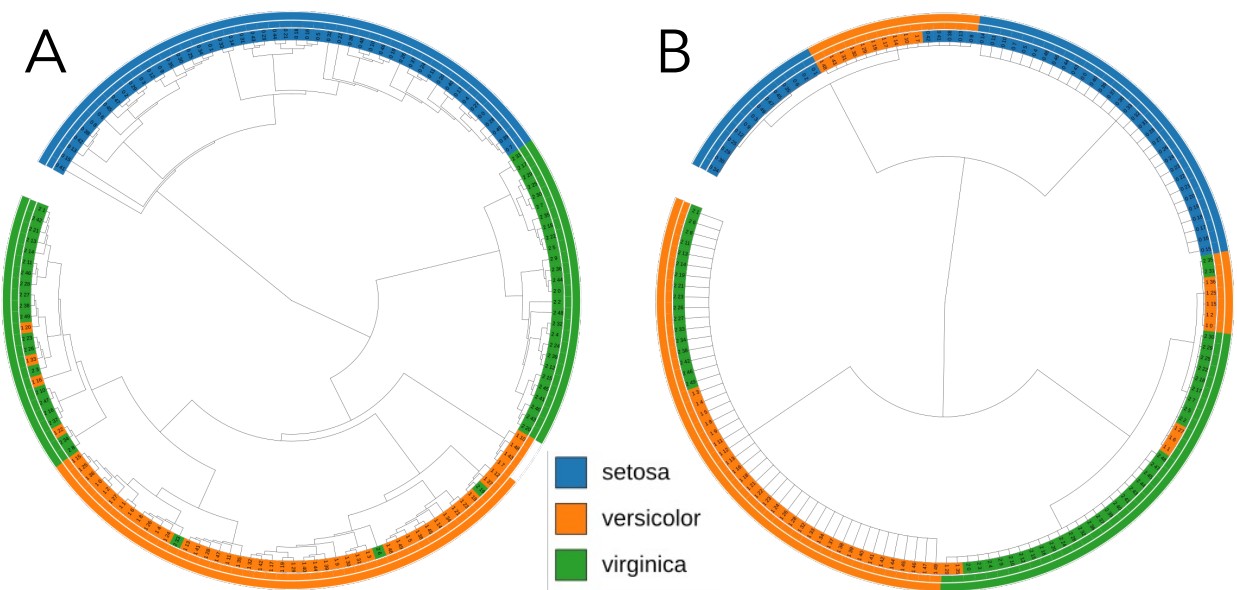

Figure S1: **Hierarchical clustering of the Iris dataset.** Comparison of hierarchical clusterings produced by the SCUT algorithm (A) and the Zha et al. (2001) algorithm (B) on the Iris dataset, which includes setosa (blue), versicolor (orange) and virginica (green) Iris flowers. In both dendograms, the three concentric rings represent the ground-truth class labels (inner ring), BU predictions (middle ring), and TD predictions (outer ring).

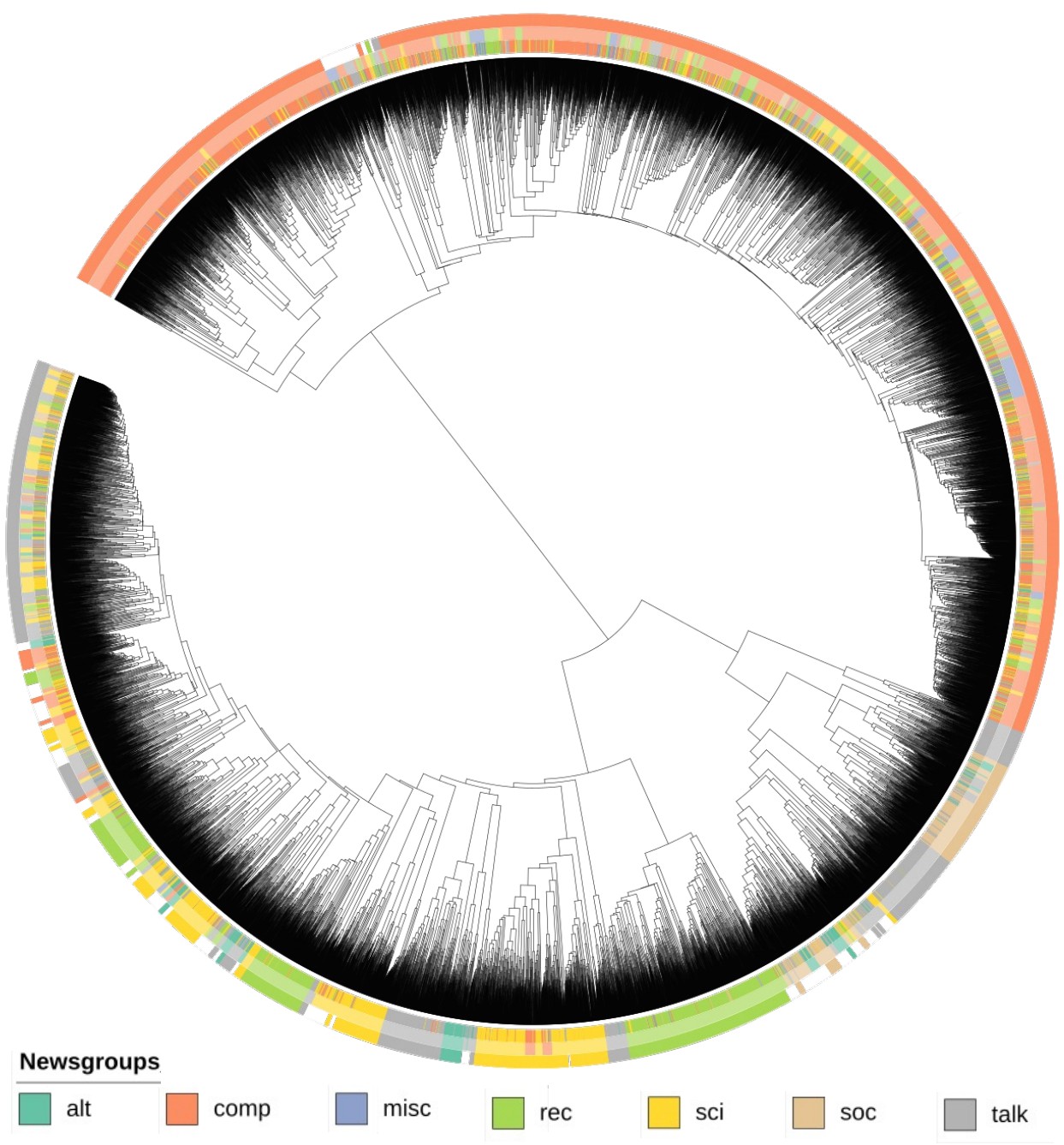

Figure S2: **Agglomerative hierarchical clustering of the 20Newsgroups dataset using Ward's metrics.** The three outer circles are colored with labels described in the inset legend. See legend in Figure 4.

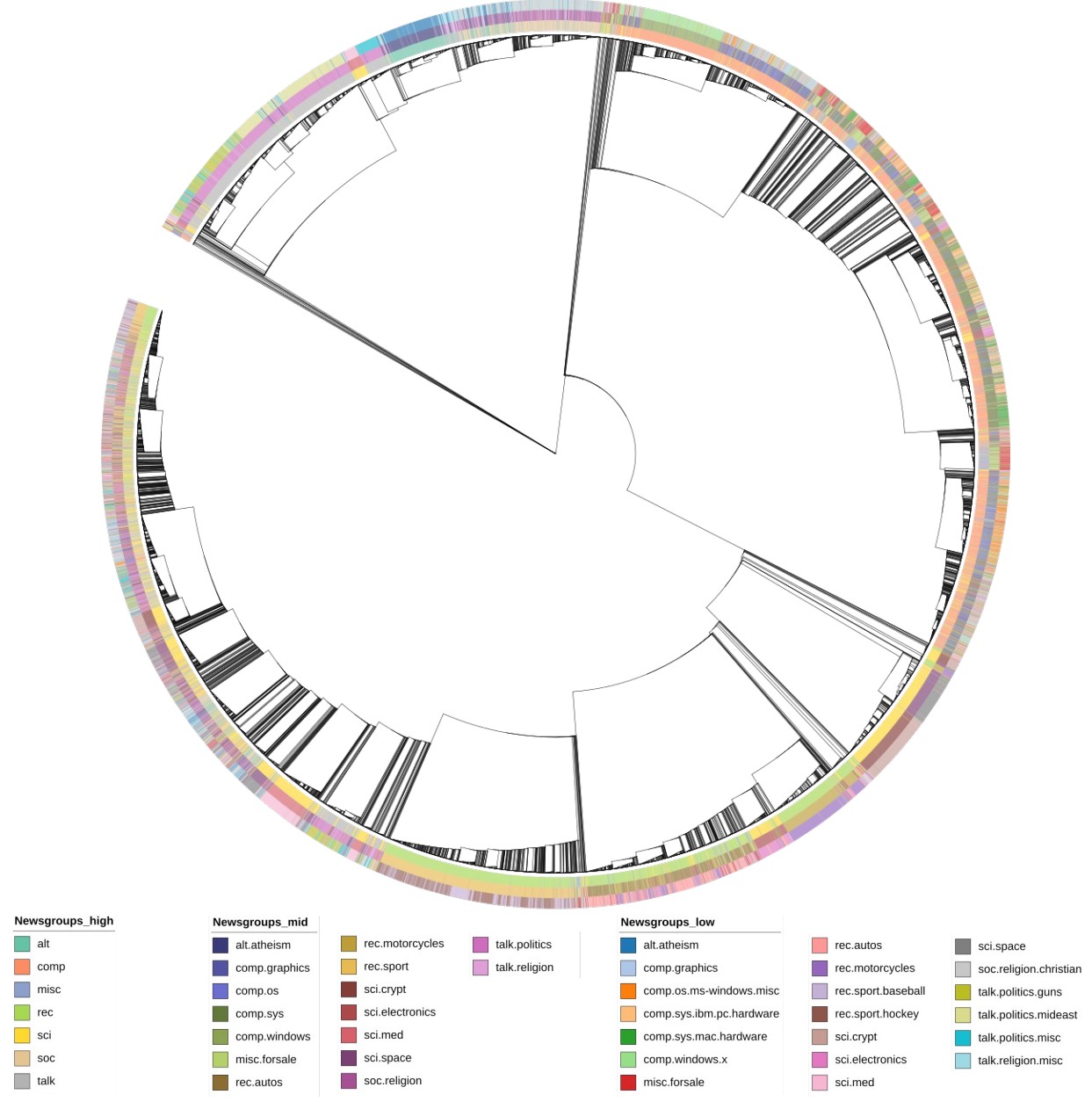

Figure S3: **SCUT dendrogram on the 20Newsgroups dataset after removing high-weight features.** The three outer circles are colored with labels described in the inset legend. See legend in Figure 4.

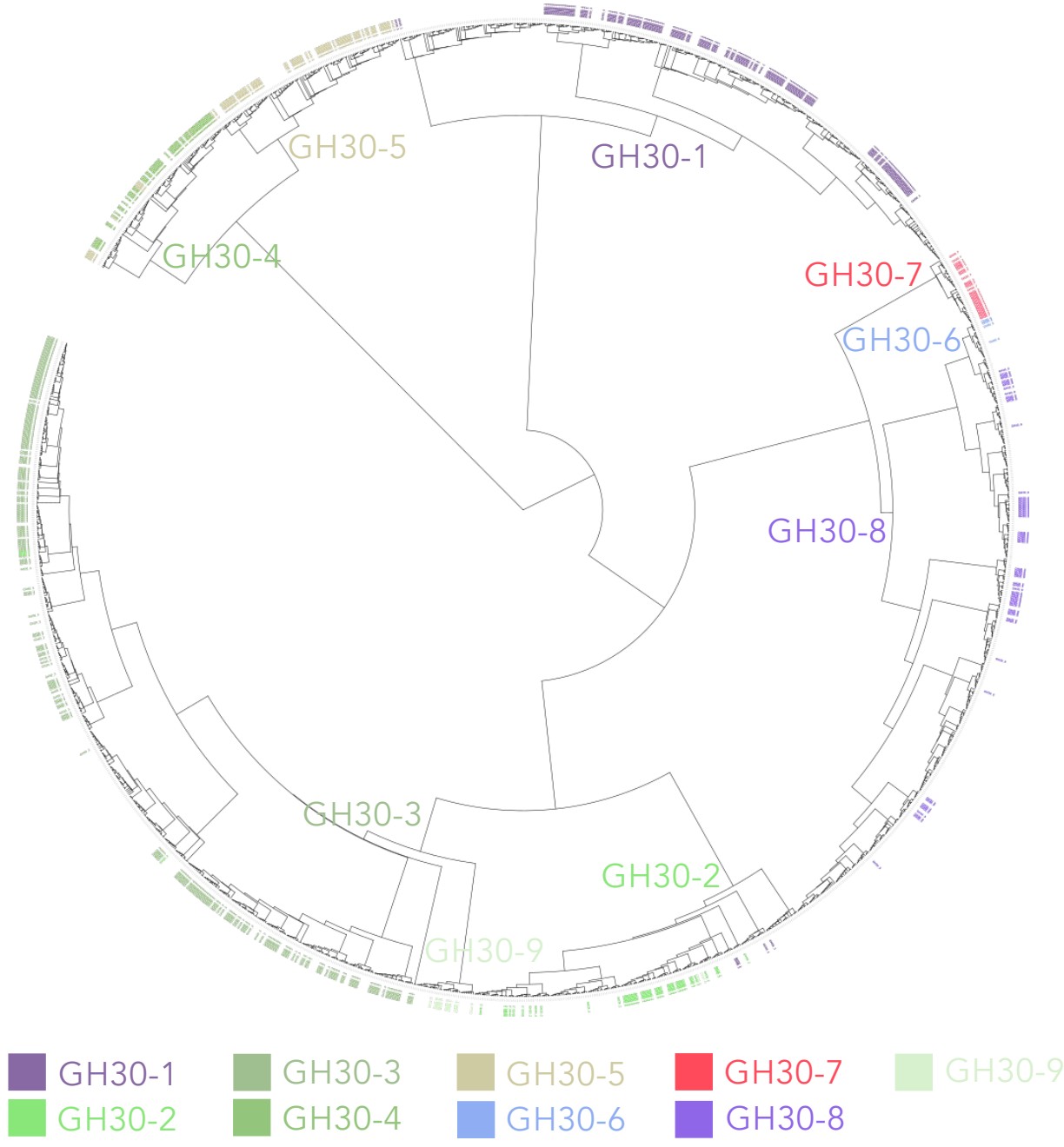

Figure S4: **GH30 dataset analysis with SCUT.** Dendrogram produced by SCUT. Leaves are annotated with the nine functional classes (GH30-1,. . . ,GH30-9) defined in the CAZy database. Subtrees predominantly containing sequences from a given class are labeled with that class name.

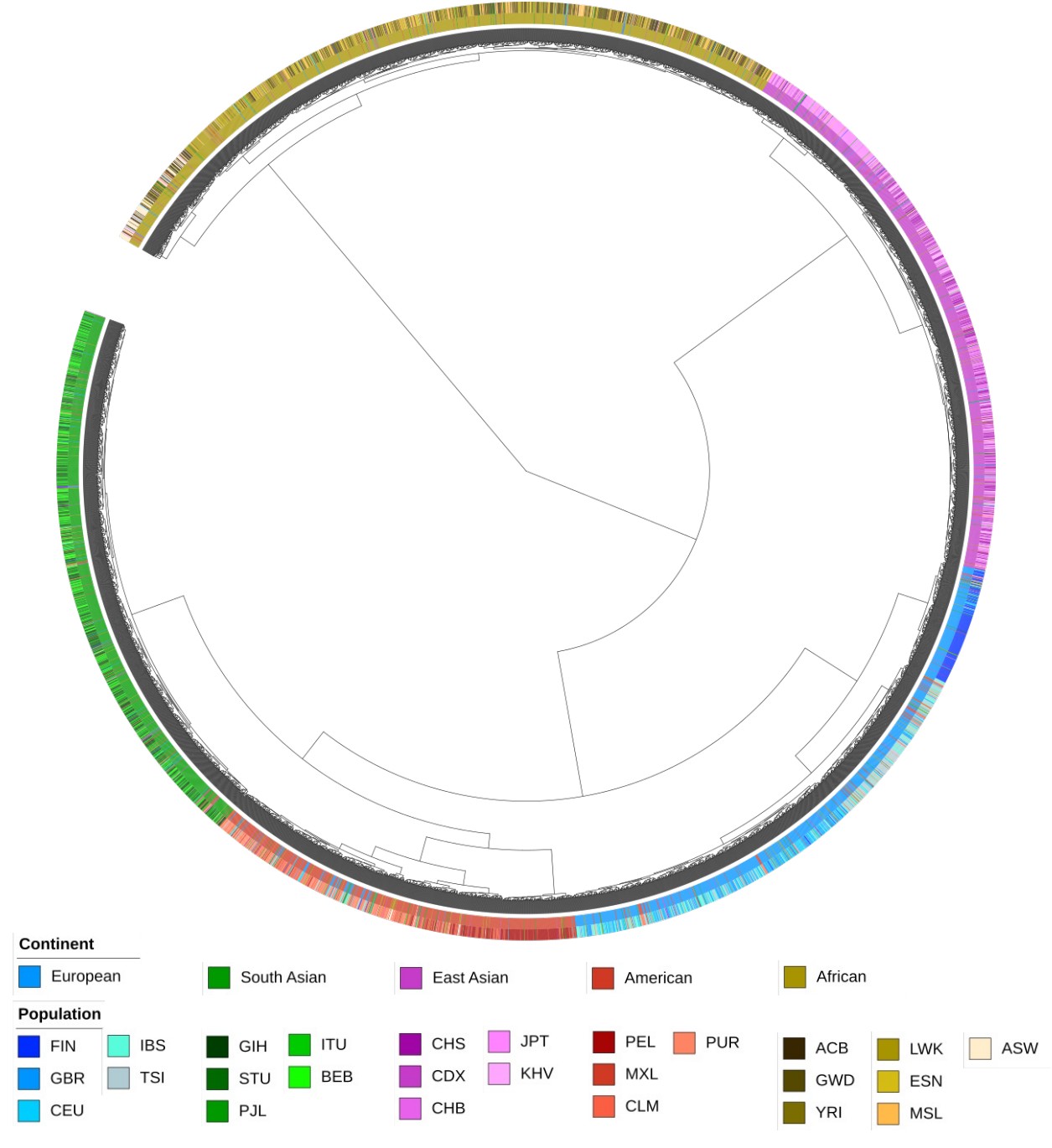

Figure S5: **Human Genome Dataset analysis with Ward on concatenated vectors.** Dendrogram produced by Ward's agglomerative clustering where individuals are represented by concatenations $(\mathbf{v}, \mathbf{1} - \mathbf{v})$. Compare to Figure 5B.

**Review of Paper6704 by Reviewer D2Ab**

Reviewby Reviewer D2Ab18 Feb 2026 at 19:03 (modified: 18 Feb 2026 at 19:05)Everyone

**Summary Of Contributions:**

The paper introduces SCUT (Spectral Clustering for Unsupervised Classification Trees), a top-down hierarchical clustering method designed for tabular data. The key idea is to model the dataset as a bipartite graph connecting observations and features, then recursively split it by approximating the Normalized Cut objective using the SVD of a normalized weight matrix.

The authors emphasize three main contributions:

1. Density-based splitting: Instead of cutting the Fiedler vector at an arbitrary threshold (like zero), SCUT treats the 1D projection as a smooth distribution and looks for natural low-density gaps between peaks. Splits happen at these "valleys," which tend to align better with meaningful structure in the data.
2. Out-of-sample prediction: At each internal node, SCUT learns and stores a linear classifier. This effectively turns the clustering procedure into an unsupervised decision tree, so new data points can be routed down the tree in logarithmic time—without recomputing the clustering from scratch.
3. Ante-hoc interpretability: By examining the feature-side singular vectors, the method measures how strongly each feature influences a split. This provides intuitive explanations for why the tree branches the way it does.

The method is evaluated on four varied datasets (Iris, French National Assembly voting, 20 Newsgroups, and Goodreads) and compared against Ward's method as well as an earlier bipartite spectral clustering approach by Zha et al. (2001).

**Are the claims made in the submission supported by accurate, convincing and clear evidence?:** No

**Explain your answer above:**

While the quantitative evaluation would benefit from some revisions, the central claims of the paper, i.e., SCUT offers a scalable, divisive hierarchical clustering framework with built-in out-of-sample inference and ante-hoc interpretability, are largely supported by solid theory.

We thank the reviewer for the positive assessment of SCUT and for recognizing that its central claims—scalability, divisive hierarchical construction, built-in out-of-sample inference, and ante-hoc interpretability—are supported by solid theoretical foundations.

We agree that the quantitative evaluation can be further strengthened. In the revised version, we have expanded the empirical analysis by adding two additional large-scale datasets from genomics: one involving allele variation across human populations and another focusing on protein sequence functional classification. These datasets provide complementary evidence of SCUT's behavior on high-dimensional biological data and illustrate its scalability and structural properties in domains where hierarchical organization is particularly relevant.

We believe that these additions strengthen the empirical support for the theoretical contributions of the paper.

In particular:

1. The use of SVD on a bipartite graph to approximate the Normalized Cut objective is mathematically sound and builds naturally on prior spectral clustering work. In Appendix A.1, the authors go further and prove that the recursive splits induce strictly convex subspaces. That result is not just a technical detail, it provides real geometric justification for why the clusters are well-structured and why the out-of-sample routing mechanism (Algorithm 2) is principled rather than ad hoc.
2. The complexity discussion in Section 3.5 is detailed and technically sound. The derivation of worst-case and best-case bounds is formalized.
3. The method appears robust across data types, as the algorithm is applied to a diverse range of datasets.

We thank the referee on this detailed assessment.

**Would at least some individuals in TMLR's audience be interested in knowing the findings of this paper?:** Yes

**Explain your answer above:**

The paper seems of possible interest to researchers and practitioners in the TMLR community, particularly those working on unsupervised learning, interpretability, and scalable methods for tabular data. Recasting hierarchical clustering as an "unsupervised decision tree" with built-in out-of-sample routing addresses a practical limitation of methods such as Ward's method, which are inherently transductive. The scalability analysis and demonstrations across multiple datasets might also suggest that the approach may be relevant across several application domains.

We thank the reviewer for his/her positive assessment of SCUT.

In line with the reviewer's observation regarding the breadth of potential applications, in the discussion, we added a paragraph indicating the applications that can be envisaged for the SCUT algorithm.

"Beyond standard benchmarks, SCUT is particularly suited to settings where the primary challenge is the structured organization of large and continuously evolving databases. In domains such as genomics, proteomics, and other large-scale scientific repositories, clustering serves not only as an exploratory tool but as a mechanism for organizing knowledge. By combining hierarchical partitioning with principled out-of-sample routing, SCUT can function as a stable organizational layer that integrates new data without requiring global recomputation. Similar requirements arise in large document collections, recommendation systems, and high-dimensional representation spaces, where scalability, interpretability, and consistent sample assignment are essential for long-term maintenance and analysis."

Following the reviewer remark, we included two additional datasets illustrating alternative applications of SCUT in genomics.

**Requested Changes:**

1. It's not clear to me that out-of-sample inference is tested. Since a major selling point of SCUT is that it can place new data points into the learned hierarchy without rebuilding the tree (Algorithm 2), it would be good to have experiments demonstrating that it works well in practice. Currently, I believe all results are transductive, as clustering is performed on the full dataset, so the paper could consider including a train/test split experiment showing how reliably the learned linear classifiers assign unseen data, ideally compared against reclustering.

    The **new section 3.8** answers to this remark.

2. Similarly, the authors introduce a custom Top-Down (TD) metric, but they explicitly state that they searched for the best threshold to maximize accuracy on the same dataset used for evaluation. Hyperparameter tuning on the target data seems artificial and would further motivate a train/test split experiment in my opinion.

    We thank the reviewer for this remark. We would like to clarify that the Top-Down (TD) procedure is not part of the SCUT model and does not influence the construction of the hierarchy. SCUT is trained entirely in an unsupervised manner, without access to labels.

    The parameter $\theta$ in the TD procedure determines how the already constructed dendrogram is mapped to label predictions for computing evaluation metrics such as ACC. Selecting $\theta$ to maximize ACC does not alter the tree structure or any model parameters; it only affects how the fixed tree is interpreted for reporting performance. This is analogous to choosing a cut level in a dendrogram to compare clusters with known labels. To avoid confusion, we revised the manuscript to clarify that TD is purely an evaluation protocol and not a model hyperparameter.

    We adapted the text **in pp11**.

**Broader Impact Concerns:**

N/A

**Review of Paper6704 by Reviewer bJv7**

Reviewby Reviewer bJv702 Feb 2026 at 06:18 (modified: 18 Feb 2026 at 19:03)Everyone

**Summary Of Contributions:**

Summary The paper suggests a hierarchical clustering algorithm that leverage normalized cuts at each level. It uses the SVD decomposition of the data matrix as proposed in Dhillon et al. 2001. The authors claim that their method works in $O(mn^2)$ in worse case complexity and $O(mn\log n)$ in the better case. They also claim that they introduce an original way to assign new samples to previously constructed tree. Authors present results on 4 data sets and claim they are of increasing difficulty. They compare with the method of Zha 2001 and ward 1963.

Strengths: 1. Some of the formal part of the algorithm are described clearly.

Weakness:

1. I find the paper minimalistic in the following aspects:

a. algorithmic design: simple, known to the SoTA construction, without novelty (although that is not the most important aspect of this journal)

The contributions of the article have been explicitly stated in the introduction now.

b. Comparison and addressing state of the art (SoTA): There is a body of work on hierarchical clustering and normalized cut. There is no dedicated related work in this paper that explain the context of this work in light of the SoTA. There are so many works that are similar including ones published in TMLR such as (Cluster Tree for Nearest Neighbor Search, Kushnir et. al.) which also uses NCUT for 1 d partitions, but in a more efficient way, Wieling et al. (2010, ACL Anthology), Zheng et al. (2018, Knowledge-Based Systems), Sugahara et al. (2023, Pattern Recognition), Ienco, Pensa, Meo (ECML PKDD 2009), and many more. Even a classical comparison with the original Spectral clsuteirng algorithm by Y Weiss, and Malik et al could be interesting in this case. This is the most problematic part in this submission: The work is presented as if it lives in a vacuum.

We thank the reviewer for pointing out the need for a clearer positioning of our work with respect to the existing literature on hierarchical clustering and Normalized Cut–based methods. We agree that the original version did not sufficiently contextualize SCUT within the broader SoTA. In the revised manuscript, we have added **a dedicated paragraph of the introduction** that discusses prior approaches to spectral clustering and normalized cut, including classical formulations (e.g., Shi & Malik; Weiss) as well as more recent hierarchical and NCUT-based methods such as Cluster Tree for Nearest Neighbor Search (Kushnir et al.) and other relevant works mentioned by the reviewer. Also, we explicitly clarify the conceptual and methodological differences between these approaches and SCUT.

Results from new comparisons are reported in **the new Table 3**.

c. Empirical studies are with the most basic methods on which the method is based on. Results are presented partially for only 3 data set. It is insufficient

We added **two new datasets** to illustrate the interest in SCUT in less classical datasets:

1. 5008 phased haplotypes coming from 2,504 individuals and characterized by 805 biallelic single-nucleotide polymorphisms (SNPs). Data is coming from the 1000 Genomes Project dataset.
2. the functional clustering of 1,675 protein sequences belonging to the glycohydrolase protein family GH30.

Comparisons with Ward agglomerative clustering are given for both datasets, based on different comparison criteria.

2. There are claims that are not established in the manuscript:

a. Author claims that the method for assigning new point is novel (alg 2). But it is one of the most basic ones in all hierarchical space partition algorithm such as RP trees, or even the above TMLR paper. This claim is quite puzzling.

We agree with the reviewer that Algorithm 2 is structurally identical to standard inference in hierarchical space-partitioning trees (e.g., RP-trees). We did not intend to claim novelty in the tree traversal mechanism itself.

Our contribution is that each spectral split yields an explicit linear decision function derived from the bipartite SVD. By storing these functions at internal nodes, SCUT supports inductive assignment of new samples without recomputing eigenvectors or rebuilding the hierarchy. This contrasts with standard spectral clustering methods, which typically require recomputation or Nyström-type extensions for out-of-sample data.

We **revised the manuscript (pp8)** to moderate the novelty claim and clarify that the contribution lies in enabling consistent out-of-sample routing within a spectral hierarchical framework.

b. The claims that there si a running time improvement over other methods is not backed by citations, or any reference to other clustering running time performance. It should also be backed by an accuracy trade-off.

To substantiate our runtime claims and clarify the accuracy–efficiency trade-off, we have added **the new Table 4**, which reports empirical running times for SCUT and the same baseline methods evaluated in **Table 3**.

When Tables 3 and 4 are considered jointly, several observations emerge:

1. **On small datasets (e.g., Iris)**, runtime differences between methods are naturally limited. In this regime, SCUT has a slightly higher computational overhead due to the SVD computation at each split. However, as shown in Table 3, SCUT achieves stronger structural and predictive performance

(MCC/ACC and dendrogram purity), indicating that the additional cost translates into improved hierarchical quality.

2. **On medium-scale datasets (French Assembly)**, SCUT remains competitive in runtime while maintaining superior structural alignment (Table 3). This suggests that the recursive spectral splitting does not introduce prohibitive computational overhead in realistic tabular settings.

3. **On larger datasets (e.g., 20Newsgroups)**, Table 4 shows that agglomerative methods such as Ward exhibit substantially higher running times, consistent with their quadratic complexity in the number of samples. In contrast, SCUT scales more favorably, in line with the theoretical analysis ($\Omega(mn \log n)$ best case, $O(mn^2)$ worst case). Importantly, Table 3 shows that this improved scaling is not achieved at the expense of structural quality: SCUT consistently matches or outperforms baselines in MCC, ACC, and dendrogram purity.

Taken together, Tables 3 and 4 demonstrate that SCUT achieves a favorable **accuracy–efficiency trade-off**: it provides improved or comparable hierarchical structure while scaling more gracefully on larger datasets.

We also note that the current implementation is not **fully** optimized (pure Python), suggesting that the observed runtime behavior represents a conservative estimate of SCUT's practical scalability. See remarks in **Section 3.9** (Complexity analysis).

These considerations are inserted in **the new Section 3.7**.

c. "in regions of high density in p are likey to correspond to meaningful clustering" – what if there are multiple such?

We agree that the projected density p of the Fiedler vector may exhibit multiple high-density regions. This situation is expected when more than two latent groups are present, and SCUT is designed to handle it naturally.

At each node, SCUT does not assume bimodality. When several peaks exist, multiple valleys may also appear. Our criterion (Eq. 16) selects the valley that maximizes the density contrast on both sides, i.e., the most structurally significant separation at that level of the hierarchy. This typically separates the data into two coherent super-groups, each possibly containing multiple modes.

Since SCUT is divisive and recursive, subsequent splits are applied independently within each branch, progressively isolating finer modes. Therefore, multiple high-density regions are not a failure case; they are precisely what enables the hierarchical structure to emerge.

Finally, when no clear valley exists (i.e., weak density contrast), the confidence score (Eq. 18) becomes small, reflecting that the split is not well supported by the data.

We adapted the text accordingly, **pp6**.

3. Paper organization and presentation is confusing, and lacking context:

Thank you for this remark that helped rewrite the abstract, introduction and the beginning of the result section.

a. Why does the abstract open with discussion on classification and data labeling while the work is on unsupervised learning?

We thank the reviewer for this remark.

The abstract has been rewritten.

The reference to classification and data labeling in the opening of the abstract was intended to motivate the practical need for structured and interpretable data organization, rather than to suggest that the proposed method is supervised. We have revised the abstract to make clearer that SCUT is a fully unsupervised framework designed to structure data in a way that can subsequently facilitate downstream tasks such as classification. We also highlighted time complexity performance to explain scalability.

b. Why are the contributions of this work mentioned in the body of the paper instead of in the introduction?

We added the list of contributions in the introduction explicitly.

c. Why are there no equation numbers for reference

We systematically added numbers to each mathematical display as requested.

d. Why is there no discussion of relevant literature in a separate section?

We added a dedicated paragraph in the introduction now. See above.

4. Empirical comparison is lacking actual comparative results: only a table with partial reporting is presented. Instead, huge illustrations are shown that don't contribute much to the understanding of each algorithm's actual performance. The data sets taken are extremely small and there is no explanation on the claim that they are of increasing difficulty: why is that so? Clearly Iris is the easiest, but how are the other different? Besides there are far more interesting and challenging data sets to work with.

   We introduced a recap table reporting the results of multiple performance measures for the different datasets. This table answers to the remark of the referee.

5. Some confusion on terminology: Cut vs cut cost function

We thank the reviewer for this remark. We now replaced in several places in the text the word "cut" with the word "split". Hopefully the ambiguities disappeared.

**Are the claims made in the submission supported by accurate, convincing and clear evidence?:** No

**Explain your answer above:**

See above a. Author claims that the method for assigning new point is novel (alg 2). But it is one of the most basic ones in all hierarchical space partition algorithm such as RP trees, or even the above TMLR paper. This claim is quite puzzling.

b. The claims that there is a running time improvement over other methods is not backed by citations, or any reference to other clustering algs running time performance. It should also be backed by an accuracy trade-off.

c. "in regions of high density in p are likey to correspond to meaningful clustering" – what if there are multiple such?

See answers above.

**Would at least some individuals in TMLR's audience be interested in knowing the findings of this paper?:** No

**Explain your answer above:**

I think the design of the algorithm is quite smilistic, uses prior construction in a hierachical manner and that bring any new angle on the matter. The Hierachical procedures are simialr to what we see in Proximity search cosntructing for search and cosntruction fo the tree. I metnioned simialr paper on that.

See answers above.

**Requested Changes:**

I think the authros need to rebuild their work in light of the background research that has been done on this method, in order to bring to light the jusitficaiton for their claims. The empirical study is also not convincing and lacking in many senses - not enough baseline to compare with, and not enough data sets.

The list of contributions is defined from the introduction now.

**Broader Impact Concerns:**

I think that the complete lack of discusison of such a large body of relevant work is problematic.

This point is now addressed.

**Review of Paper6704 by Reviewer e1wy**

Reviewby Reviewer e1wy 11 Jan 2026 at 11:31 (modified: 18 Feb 2026 at 19:03)
Everyone

**Summary Of Contributions:**

Summary of contributions

The paper proposes SCUT, a top-down (divisive) hierarchical clustering method for nonnegative observation-by-feature matrices that models the data as a bipartite graph between observations and features. At each node, it performs a bipartite spectral 2-way split by computing a truncated SVD of a normalized matrix , uses the second left singular vector as a 1D embedding of observations, and selects a split threshold via a density-valley criterion rather than a fixed sign cut. The method recursively builds a binary clustering tree down to singleton leaves.

A key additional contribution is an explicit out-of-sample assignment mechanism: each internal node stores a linear decision rule derived from the SVD (feature-side singular vector, normalization, singular value, and learned threshold), enabling fast routing of new samples through the fixed hierarchy similar to decision tree inference. The paper also emphasizes interpretability via (i) path-based explanations through the stored split rules, (ii) feature-level contribution scores for a given decision using the bipartite spectral quantities, and (iii) a geometric claim that node regions are convex in the space induced by the method's normalization. Experiments on Iris, 20 Newsgroups, and French National Assembly voting show improved leaf-level label agreement (ACC, MCC under their BU/TD protocols) compared to a prior bipartite spectral baseline, and on 20NG also compared to Ward, with a qualitative case study on Goodreads.

Key strengths

· Clear algorithmic idea that combines bipartite spectral splits with a recursive divisive hierarchy and an explicit inference procedure for new points (practical for deployment settings where new samples arrive).

· The density-valley thresholding is a reasonable attempt to stabilize splits compared to naive sign thresholding of the spectral vector.

· Interpretability story is coherent for linear splits: (i) transparent decision path, (ii) per-feature contribution decomposition of the projection score, (iii) convexity property follows naturally from half-space intersections (in the transformed space).

· Strong empirical improvements over the main baseline (Zha et al.) across reported datasets using their BU/TD evaluation, and competitive performance against Ward on 20 Newsgroups.

· Complexity discussion is explicit, including best/worst case and a proposed balancing heuristic to enforce  worst-case behavior.

We thank the reviewer for the positive assessment of SCUT and for recognizing that its central claims—scalability, divisive hierarchical construction, built-in out-of-sample inference, and ante-hoc interpretability—are supported by solid theoretical foundations.

Key weaknesses / limitations

· Evaluation primarily reduces the hierarchy to per-leaf label predictions (BU/TD) and reports ACC/MCC; there is no direct quantitative comparison of hierarchical structure quality (for example, dendrogram purity, hierarchical F-measure, cophenetic correlation, tree edit distances, or comparisons at multiple cut levels).

We introduced a number of **new evaluations** of SCUT reported in new tables:

- **Table 3** reports performances of SCUT and other baseline clustering algorithms which is based on dendrogram purity scores.
- **Table 4** reports a computational time comparison of the same clustering methods considered in Table 3.
- **Table 5** and **Table 6** report the stability of SCUT in the introduction of new points in the tree/space and its precision, based on dendrogram purity, in out-of-sample insertion.

· The TD protocol tunes a threshold to maximize accuracy, and the outlier handling depends on a user-chosen ; this introduces dataset- and label-dependent tuning that may not reflect purely unsupervised performance, and it complicates fair comparison unless all methods are tuned identically.

We thank the reviewer for raising this important point. We would like to clarify that both TD and BU are **post-hoc evaluation protocols** applied after the dendrogram is fully constructed; no label information is used at any stage of SCUT's training or tree construction. The threshold θ in TD and the minimal subtree size k are used solely to map a fixed hierarchy to flat label predictions for the purpose of computing ACC/MCC, which is an external validation step common to hierarchical clustering evaluation. Importantly, the same TD/BU procedures, including θ selection and the choice of k, were applied identically across all compared methods, ensuring a fair comparison. To further address the reviewer's concern, we clarified this distinction in the manuscript and additionally report hierarchical metrics that do not rely on any label-dependent tuning (e.g., dendrogram purity and multi-level cut evaluation), confirming that SCUT's performance advantage does not depend on the TD protocol.

· The claimed novelty of out-of-sample insertion is incremental in the broader literature (out-of-sample extensions and incremental clustering are known); the paper's novelty is more in the specific SVD-derived per-node rule for this bipartite divisive construction, but the positioning versus related work could be strengthened.

The positioning of SCUT wrt the operation of out-of-sample introduction has been explained explicitly in the text now. Previous work on out-of-sample insertion has been cited.

· Worst-case training complexity remains without the balancing modification, and the paper notes the current Python implementation is slower than

optimized hierarchical clustering implementations, so scalability claims may depend on implementation details and split balance.

Thank you for raising this point. We have significantly optimized the Python implementation of SCUT and updated the manuscript accordingly. First, the statement referring to a "slow implementation" in Section 4 has been removed. The optimization steps are now described in **Appendix A4**, and **Section 4** has been rewritten to reference this appendix. The main improvements are:

● Cherry-case optimization: small subproblems occurring at the end of recursion (e.g., clusters of size two) are now handled explicitly, avoiding unnecessary SVD computations.
● Reduced memory usage: the previous implementation stored multiple copies of the data matrix during recursion, effectively doubling memory consumption. The revised implementation keeps only the main matrix and creates temporary local matrices for SVD computations, which are released immediately afterward.
● FFT-based KDE: since threshold selection operates on the one-dimensional Fiedler vector and requires density evaluation on evenly spaced points, we now compute the KDE using FFT-based convolution, substantially accelerating this step.

These improvements reduce both runtime and memory overhead in practice while leaving the theoretical complexity analysis unchanged. The worst-case complexity without the balancing modification remains $O(mn^2)$, while the balancing modification still guarantees $O(mn\log n)$.

· The method relies on linear (affine) cuts at each node, which limits representational power for non-convex cluster shapes, potentially requiring many splits to approximate complex geometries.

We agree that each SCUT split corresponds to a linear decision function, and therefore each node defines a convex subspace in the transformed space $T(I)$. This is an intentional design choice. SCUT constructs a hierarchical partition composed of convex regions, which provides several advantages: geometric coherence, interpretability of splits, stability under interpolation, and efficient inductive routing of new samples.

While individual splits are linear, the recursive composition of such splits yields a piecewise-linear decision boundary. As in classical decision trees, hierarchical combinations of linear cuts can approximate complex and non-convex structures to arbitrary precision, at the cost of additional depth. Thus, SCUT does not fundamentally restrict representational capacity, but rather controls it through hierarchical refinement.

Importantly, our objective differs from that of flat spectral clustering methods that directly recover non-convex clusters without providing an inductive

structure. SCUT explicitly trades global nonlinear partitioning for a structured, interpretable, and scalable hierarchy with native out-of-sample assignment. This tradeoff is deliberate and aligned with the goal of constructing unsupervised decision trees.

We have clarified this point in the revised manuscript, **pp 9 and pp22**.

· Threshold selection via KDE/grid search may be sensitive to bandwidth/grid choices, but robustness analyses or ablations around these hyperparameters are limited.

We thank the reviewer for raising this point. To assess the sensitivity of the KDE-based threshold selection to the bandwidth parameter, we performed a bandwidth ablation analysis reported in **Appendix A.3 (Figure S1)**. In this experiment, we varied the bandwidth factor bf used to scale Silverman's rule-of-thumb bandwidth $h=1.06\sigma^{\wedge}n{-}1/5$ and evaluated the resulting SCUT trees using dendrogram purity on the Human Genome Dataset at both the continent and population levels.

The results show that performance remains stable across a broad range of bandwidth factors, with a shallow optimum around bf≈0.5, which corresponds to the default value used in SCUT. Importantly, deviations from this value lead only to minor variations in purity, indicating that the KDE-based thresholding procedure is not overly sensitive to the precise bandwidth choice.

We have clarified this robustness analysis in **Appendix A.3**.

· The method assumes (or enforces via preprocessing) nonnegative features; applicability to arbitrary real-valued features may require additional discussion or preprocessing choices that could affect results.

We thank the reviewer for highlighting this point. SCUT operates on the bipartite representation of the data matrix and therefore assumes non-negative entries, following the formulation used in bipartite spectral clustering methods such as \cite{Dhillon2001,Zha2001}. In practice, this requirement is not restrictive, as arbitrary real-valued features can be transformed into non-negative form through simple preprocessing steps such as min–max scaling, feature shifting, or sigmoid transformations. These transformations preserve the relative structure of the data while ensuring compatibility with the bipartite spectral framework.

In our experiments, we applied such preprocessing when necessary (e.g., normalization or complement features). We clarified in **Section 2.3** that SCUT operates on non-negative tabular data, following the standard bipartite spectral clustering formulation \citep{Dhillon_clust,Zha_Clust}, and that simple preprocessing such as min–max scaling or sigmoid transformations can be used when real-valued features are present.

**Are the claims made in the submission supported by accurate, convincing and clear evidence?:** No

**Explain your answer above:**

The paper provides clear algorithmic descriptions (Algorithms 1 and 2), a formal proof for the convexity property, and quantitative results (ACC/MCC under BU and TD) showing consistent improvements over the main bipartite spectral baseline (and Ward on 20 Newsgroups), so the core claim that SCUT can produce hierarchies that better align with class labels on the reported benchmarks is reasonably supported. However, several broader claims are only partially evidenced: the density-valley thresholding is motivated intuitively but lacks ablations against simpler threshold rules, the out-of-sample insertion mechanism is described but not evaluated in a targeted way (quality or runtime versus rebuilding), and interpretability is demonstrated via a small number of illustrative cases (for example, showing sensible "influential terms" for one 20 Newsgroups split) rather than showing that these feature-attribution explanations are consistently stable and informative across datasets, tree levels, and branches. Overall, the evidence is solid for the method definition and the reported benchmark improvements, but incomplete for the stronger claims about why it works, insertion benefits, and the breadth of the interpretability story.

See above, the point-by point answers.

**Would at least some individuals in TMLR's audience be interested in knowing the findings of this paper?:** Yes

**Explain your answer above:**

Yes. The paper targets an intersection of topics that many in TMLR's audience care about, namely scalable clustering, spectral methods, and interpretability, and it proposes a divisive hierarchical variant with an explicit out-of-sample routing mechanism. Readers interested in spectral clustering on large sparse matrices (for example, text or interaction data), hierarchical representations of unlabeled data, and practical deployment issues (assigning new points without recomputing a full clustering) would likely find the approach and empirical results relevant, even if additional evaluation is needed to fully substantiate the broader claims.

See above, the point-by point answers.

**Requested Changes:**

Critical

1. **Add a direct evaluation of the hierarchical structure (not only leaf-level label agreement).**
   Include at least one standard hierarchy-aware metric (for example, dendrogram purity, hierarchical F-measure, cophenetic correlation, or a family of scores over multiple cut levels) and compare against the same baselines. Current BU/TD + ACC/MCC primarily evaluates induced label predictions and does not directly assess whether the learned tree is structurally better.

   We thank the reviewer for this suggestion. To directly assess the structural quality of the learned hierarchies, we have now included **dendrogram purity**, which evaluates how well the tree topology aligns with ground-truth class partitions and does not require a distance matrix or hierarchical labels.

   Regarding the other suggested metrics:

   ● **Cophenetic correlation** measures how faithfully a dendrogram preserves an underlying pairwise distance matrix. Since SCUT does not construct the hierarchy by progressively approximating a predefined dissimilarity matrix (as in agglomerative clustering), but instead performs recursive spectral bipartitions, there is no reference distance structure for such a correlation to evaluate.
   ● **Hierarchical F-measure** requires ground-truth hierarchical labels. While such a metric could in principle be applied to datasets like 20Newsgroups or HGD, where some hierarchical structure can be inferred from the label taxonomy, most datasets considered in our experiments provide only flat annotations. Even in the cases where a hierarchy exists, applying hierarchical F-measure would require defining post-hoc strategies to map the learned clusters to the reference hierarchy, introducing additional modeling choices that could affect the evaluation.

   For these reasons, dendrogram purity provides the most consistent hierarchy-aware metric across all datasets considered in this study. We therefore report this metric and compare it with the same baselines accordingly (see **Table 3**).

2. **Ablate the thresholding rule.**
   Quantify the impact of the density-valley thresholding by comparing against simpler alternatives (for example, sign threshold at 0, median split, quantile-based split, or 1D k-means on ) under the same experimental protocol. Without this, it is unclear whether gains come from the thresholding innovation or from the underlying bipartite spectral splits alone.

   Thank you for this suggestion. We agree that it is important to isolate the contribution of the thresholding rule.

In fact, **Table 3** includes several ablations of the thresholding strategy within SCUT. In addition to the proposed KDE-based thresholding (`scut_kde`), we evaluated three simpler alternatives under the same experimental protocol:

- `scut_zero`: the split is performed at 0, corresponding to the standard sign-based partitioning commonly used in spectral clustering.
- `scut_median`: the split is performed at the median of the Fiedler vector, producing balanced partitions independently of the density structure.
- `scut_kmeans++`: the threshold is obtained by applying 1D k-means clustering (k=2) on the Fiedler vector and placing the threshold between the two centroids.

These variants isolate the effect of the thresholding rule while keeping the rest of the SCUT pipeline unchanged. As shown in **Table 3**, the KDE-based strategy (`scut_kde`) generally achieves stronger or more stable dendrogram purity across datasets, particularly on structured datasets such as the French National Assembly and the Human Genome Dataset. Simpler thresholding rules tend to degrade performance, especially in cases where the projected distribution exhibits multiple modes or asymmetric density structure. These results suggest that the performance gains are not solely due to the underlying bipartite spectral split, but also benefit from the density-aware threshold selection.

In addition, we conducted a **robustness analysis** of the KDE-based thresholding with respect to the **bandwidth parameter.** As mentioned above, to assess the sensitivity of the KDE-based threshold selection to the bandwidth parameter, we performed a bandwidth ablation analysis reported in **Appendix A.3 (Figure S1)**. In this experiment, we varied the bandwidth factor bf used to scale Silverman's rule-of-thumb bandwidth $h=1.06\sigma n^{-1/5}$ and evaluated the resulting SCUT trees using dendrogram purity on the Human Genome Dataset at both the continent and population levels.

The results show that performance remains stable across a broad range of bandwidth factors, with a shallow optimum around bf≈0.5, which corresponds to the default value used in SCUT. Importantly, deviations from this value lead only to minor variations in purity, indicating that the KDE-based thresholding procedure is not overly sensitive to the precise bandwidth choice. We have clarified this robustness analysis in **Appendix A.3**.

3. **Substantiate the "out-of-sample insertion" advantage with experiments.**
   Evaluate insertion on held-out data: build the tree on a training set, insert test points using Algorithm 2, and measure (i) label agreement, (ii) consistency with rebuilding on full data, and (iii) runtime versus rebuilding or versus other

incremental/extension baselines. The mechanism is described, but its practical benefit is not demonstrated.

This has been done. See above, and new **Table 5** and **Table 6**.

Non-critical (would strengthen the paper)

4. **Broaden and modernize baselines.**
   Add stronger and more diverse comparisons (for example, average linkage/complete linkage, bisecting k-means, scalable spectral variants, and one or two hierarchical topic/document clustering baselines for 20NG). The current baseline set is limited (Zha throughout, Ward only on 20NG).

See answer above and analysis reported in **Section 3.7** and **Table 3**.

5. **Report runtime and memory benchmarks.**
   Provide end-to-end wall-clock and memory usage comparisons for training and (especially) inference/insertion, including the effect of split balance and the proposed balancing modification.

Runtime has been measured against other algorithms and reported in **Table 4**.

We improved the implementation of the SCUT algorithm, also improving memory usage. A section on memory complexity has been added to Section 3.9 "Complexity analysis".

6. **Clarify and tighten claims and terminology.**
   Resolve any confusing statements around convexity/linearity limitations and ensure novelty claims (especially for insertion and interpretability) are positioned carefully relative to prior work.

The text has been improved in many ways and unclear points are hopefully resolved.

7. **Improve experimental reporting rigor.**
   Add variance estimates (multiple runs where randomness exists), sensitivity analysis for and in TD, and clearer justification of hyperparameter choices (for example KDE bandwidth/grid settings for threshold selection).

We performed **new analyses (bandwidth)**. See new tests on bandwidth settings.

8. **Strengthen interpretability evaluation.**
   Beyond examples, include basic sanity checks for explanation faithfulness and stability (for example, perturb/remove top-ranked features and observe branch changes, or measure stability of top features across resamples) and provide examples on more than one dataset/domain.

We have added several analyses to address the suggestion to strengthen the evaluation of interpretability.

**Explanation stability.**
SCUT relies on spectral cuts defined by the Fiedler vector of the graph Laplacian. When the second-smallest eigenvalue is simple, the Fiedler vector is unique (up to

sign), which makes the corresponding split uniquely defined. Non-uniqueness occurs only in the presence of strong graph symmetries or multiple equally optimal spectral cuts, situations that are highly unlikely in real datasets. In our implementation, if such a case occurs, one of the equivalent solutions is selected. Therefore the explanation associated with each split is theoretically stable. In practice, minor variations may arise from the numerical eigensolver used to compute the Fiedler vector, but these reflect numerical optimization effects rather than instability of the SCUT methodology. We added a short sentence in the **Conclusions** with this observation.

**Explanation faithfulness.**
We added a **feature ablation experiment** on the 20Newsgroups dataset (Section 3.3, p.17 and new **Figure S3**). We removed the highest-weight features associated with the sport cluster and recomputed the dendrogram. After removal, the rec.sport.hockey and rec.sport.baseball documents become mixed, confirming that the top-ranked features identified by SCUT play a causal role in separating these subclusters.

**Stability across resampled data.**
To further evaluate robustness, **Table 5** now reports a 5-fold cross-validation experiment using out-of-sample insertion on the HGD dataset. The resulting variations in accuracy and MCC remain small across folds, indicating that the clustering structure and associated explanations are stable under sampling variability.

Together, these additions provide empirical evidence supporting both the **faithfulness and stability of SCUT explanations across multiple datasets**, complementing the qualitative examples already presented in the manuscript.

**Broader Impact Concerns:**

None.