# OpenReview forum: "SCUT : Spectral Clustering for Unsupervised classification Trees"
_TMLR — Rejected by TMLR_

### Review · Reviewer_e1wy · 2026-01-11

**Summary Of Contributions:**

## Summary of contributions

The paper proposes SCUT, a top-down (divisive) hierarchical clustering method for nonnegative observation-by-feature matrices that models the data as a bipartite graph between observations and features. At each node, it performs a bipartite spectral 2-way split by computing a truncated SVD of a normalized matrix $W_N = D_1^{-1/2} W D_2^{-1/2}$, uses the second left singular vector as a 1D embedding of observations, and selects a split threshold via a density-valley criterion rather than a fixed sign cut. The method recursively builds a binary clustering tree down to singleton leaves.

A key additional contribution is an explicit out-of-sample assignment mechanism: each internal node stores a linear decision rule derived from the SVD (feature-side singular vector, normalization, singular value, and learned threshold), enabling fast routing of new samples through the fixed hierarchy similar to decision tree inference. The paper also emphasizes interpretability via (i) path-based explanations through the stored split rules, (ii) feature-level contribution scores for a given decision using the bipartite spectral quantities, and (iii) a geometric claim that node regions are convex in the space induced by the method’s normalization. Experiments on Iris, 20 Newsgroups, and French National Assembly voting show improved leaf-level label agreement (ACC, MCC under their BU/TD protocols) compared to a prior bipartite spectral baseline, and on 20NG also compared to Ward, with a qualitative case study on Goodreads.

## Key strengths

- Clear algorithmic idea that combines bipartite spectral splits with a recursive divisive hierarchy and an explicit inference procedure for new points (practical for deployment settings where new samples arrive).
- The density-valley thresholding is a reasonable attempt to stabilize splits compared to naive sign thresholding of the spectral vector.
- Interpretability story is coherent for linear splits: (i) transparent decision path, (ii) per-feature contribution decomposition of the projection score, (iii) convexity property follows naturally from half-space intersections (in the transformed space).
- Strong empirical improvements over the main baseline (Zha et al.) across reported datasets using their BU/TD evaluation, and competitive performance against Ward on 20 Newsgroups.
- Complexity discussion is explicit, including best/worst case and a proposed balancing heuristic to enforce $O(mn\log n)$ worst-case behavior.

## Key weaknesses / limitations

- Evaluation primarily reduces the hierarchy to per-leaf label predictions (BU/TD) and reports ACC/MCC; there is no direct quantitative comparison of hierarchical structure quality (for example, dendrogram purity, hierarchical F-measure, cophenetic correlation, tree edit distances, or comparisons at multiple cut levels).
- The TD protocol tunes a threshold $\theta$ to maximize accuracy, and the outlier handling depends on a user-chosen $k$; this introduces dataset- and label-dependent tuning that may not reflect purely unsupervised performance, and it complicates fair comparison unless all methods are tuned identically.
- The claimed novelty of out-of-sample insertion is incremental in the broader literature (out-of-sample extensions and incremental clustering are known); the paper’s novelty is more in the specific SVD-derived per-node rule for this bipartite divisive construction, but the positioning versus related work could be strengthened.
- Worst-case training complexity remains $O(mn^2)$ without the balancing modification, and the paper notes the current Python implementation is slower than optimized hierarchical clustering implementations, so scalability claims may depend on implementation details and split balance.
- The method relies on linear (affine) cuts at each node, which limits representational power for non-convex cluster shapes, potentially requiring many splits to approximate complex geometries.
- Threshold selection via KDE/grid search may be sensitive to bandwidth/grid choices, but robustness analyses or ablations around these hyperparameters are limited.
- The method assumes (or enforces via preprocessing) nonnegative features; applicability to arbitrary real-valued features may require additional discussion or preprocessing choices that could affect results.

**Audience:**

Yes

**Audience Explanation:**

Yes. The paper targets an intersection of topics that many in TMLR’s audience care about, namely scalable clustering, spectral methods, and interpretability, and it proposes a divisive hierarchical variant with an explicit out-of-sample routing mechanism. Readers interested in spectral clustering on large sparse matrices (for example, text or interaction data), hierarchical representations of unlabeled data, and practical deployment issues (assigning new points without recomputing a full clustering) would likely find the approach and empirical results relevant, even if additional evaluation is needed to fully substantiate the broader claims.

**Broader Impact Concerns:**

None.

**Claims And Evidence:**

No

**Claims Explanation:**

The paper provides clear algorithmic descriptions (Algorithms 1 and 2), a formal proof for the convexity property, and quantitative results (ACC/MCC under BU and TD) showing consistent improvements over the main bipartite spectral baseline (and Ward on 20 Newsgroups), so the core claim that SCUT can produce hierarchies that better align with class labels on the reported benchmarks is reasonably supported. However, several broader claims are only partially evidenced: the density-valley thresholding is motivated intuitively but lacks ablations against simpler threshold rules, the out-of-sample insertion mechanism is described but not evaluated in a targeted way (quality or runtime versus rebuilding), and interpretability is demonstrated via a small number of illustrative cases (for example, showing sensible “influential terms” for one 20 Newsgroups split) rather than showing that these feature-attribution explanations are consistently stable and informative across datasets, tree levels, and branches. Overall, the evidence is solid for the method definition and the reported benchmark improvements, but incomplete for the stronger claims about why it works, insertion benefits, and the breadth of the interpretability story.

**Requested Changes:**

## Requested Changes

### Critical

1. **Add a direct evaluation of the hierarchical structure (not only leaf-level label agreement).**
   Include at least one standard hierarchy-aware metric (for example, dendrogram purity, hierarchical F-measure, cophenetic correlation, or a family of scores over multiple cut levels) and compare against the same baselines. Current BU/TD + ACC/MCC primarily evaluates induced label predictions and does not directly assess whether the learned tree is structurally better.

2. **Ablate the thresholding rule.**
   Quantify the impact of the density-valley thresholding by comparing against simpler alternatives (for example, sign threshold at 0, median split, quantile-based split, or 1D k-means on $f_u$) under the same experimental protocol. Without this, it is unclear whether gains come from the thresholding innovation or from the underlying bipartite spectral splits alone.

3. **Substantiate the “out-of-sample insertion” advantage with experiments.**
   Evaluate insertion on held-out data: build the tree on a training set, insert test points using Algorithm 2, and measure (i) label agreement, (ii) consistency with rebuilding on full data, and (iii) runtime versus rebuilding or versus other incremental/extension baselines. The mechanism is described, but its practical benefit is not demonstrated.

### Non-critical (would strengthen the paper)

4. **Broaden and modernize baselines.**
   Add stronger and more diverse comparisons (for example, average linkage/complete linkage, bisecting k-means, scalable spectral variants, and one or two hierarchical topic/document clustering baselines for 20NG). The current baseline set is limited (Zha throughout, Ward only on 20NG).

5. **Report runtime and memory benchmarks.**
   Provide end-to-end wall-clock and memory usage comparisons for training and (especially) inference/insertion, including the effect of split balance and the proposed balancing modification.

6. **Clarify and tighten claims and terminology.**
   Resolve any confusing statements around convexity/linearity limitations and ensure novelty claims (especially for insertion and interpretability) are positioned carefully relative to prior work.

7. **Improve experimental reporting rigor.**
   Add variance estimates (multiple runs where randomness exists), sensitivity analysis for $k$ and $\theta$ in TD, and clearer justification of hyperparameter choices (for example KDE bandwidth/grid settings for threshold selection).

8. **Strengthen interpretability evaluation.**
   Beyond examples, include basic sanity checks for explanation faithfulness and stability (for example, perturb/remove top-ranked features and observe branch changes, or measure stability of top features across resamples) and provide examples on more than one dataset/domain.

---

### Review · Reviewer_bJv7 · 2026-02-02

**Summary Of Contributions:**

Summary
The paper suggests a hierarchical clustering algorithm that leverage normalized cuts at each level. It uses the SVD decomposition of the data matrix as proposed in Dhillon et al. 2001. The authors claim that their method works in O(mn^2) in worse case complexity and O(mnlogn) in the better case. They also claim that they introduce an original way to assign new samples to previously constructed tree. Authors present results on 4 data sets and claim they are of increasing difficulty. They compare with the method of Zha 2001 and ward 1963.

Strengths:
1.	Some of the formal part of the algorithm are described clearly.
Weakness:

1.	I find the paper minimalistic in the following aspects:

a.	 algorithmic design: simple, known to the SoTA construction, without novelty (although that is not the most important aspect of this journal)

b.	Comparison and addressing state of the art (SoTA): There is a body of work on hierarchical clustering and normalized cut. There is no dedicated related work in this paper that explain the context of this work in light of the SoTA. There are so many works that are similar including ones published in TMLR such as (Cluster Tree for Nearest Neighbor Search, Kushnir et. al.) which also uses NCUT for 1 d partitions, but in a more efficient way, Wieling et al. (2010, ACL Anthology),  Zheng et al. (2018, Knowledge-Based Systems), Sugahara et al. (2023, Pattern Recognition), Ienco, Pensa, Meo (ECML PKDD 2009), and many more.
Even a classical comparison with the original Spectral clsuteirng algorithm by Y Weiss, and Malik et al could be interesting in this case.
		This is the most problematic part in this submission: The work is presented as if it lives in a vacuum.

c.	Empirical studies are with the most basic methods on which the method is based on. Results are presented partially for only 3 data set. It is insufficient

2.	There are claims that are not established in the manuscript:

a.	Author claims that the method for assigning new point is novel (alg 2). But it is one of the most basic ones in all hierarchical space partition algorithm such as RP trees, or even the above TMLR paper. This claim is quite puzzling.

b.	The claims that there si a running time improvement over other methods is not backed by citations, or any reference to other clustering running time performance. It should also be backed by an accuracy trade-off.

c.	“in regions of high density in p are likey to correspond to meaningful clustering” – what if there are multiple such?

3.	Paper organization and presentation is confusing, and lacking context:

a.	Why does the abstract open with discussion on classification and data labeling while the work is on unsupervised learning?

b.	Why are the contributions of this work mentioned in the body of the paper instead of in the introduction?

c.	Why are there no equation numbers for reference

d.	Why is there no discussion of relevant literature in a separate section?

4.	Empirical comparison is lacking actual comparative results: only a table with partial reporting is presented. Instead, huge illustrations are shown that don’t contribute much to the understanding of each algorithm’s actual performance. The data sets taken are extremely small and there is no explanation on the claim that they are of increasing difficulty: why is that so? Clearly Iris is the easiest, but how are the other different? Besides there are far more interesting and challenging data sets to work with.

5.	Some confusion on terminology:
Cut vs cut cost function

**Audience:**

No

**Audience Explanation:**

I think the design of the algorithm is quite smilistic, uses prior construction in a hierachical manner and that bring any new angle on the matter. The Hierachical procedures are simialr to what we see in Proximity search cosntructing for search and cosntruction fo the tree. I metnioned simialr paper on that.

**Broader Impact Concerns:**

I think that the complete lack of discusison of such a large body of relevant work is problematic.

**Claims And Evidence:**

No

**Claims Explanation:**

See above
a. Author claims that the method for assigning new point is novel (alg 2). But it is one of the most basic ones in all hierarchical space partition algorithm such as RP trees, or even the above TMLR paper. This claim is quite puzzling.

b. The claims that there is a running time improvement over other methods is not backed by citations, or any reference to other clustering algs running time performance. It should also be backed by an accuracy trade-off.

c. “in regions of high density in p are likey to correspond to meaningful clustering” – what if there are multiple such?

**Requested Changes:**

I think the authros need to rebuild their work in light of the background research that has been done on this method, in order to bring to light the jusitficaiton for their claims. The empirical study is also not convincing and lacking in many senses - not enough baseline to compare with, and not enough data sets.

---

### Review · Reviewer_D2Ab · 2026-02-18

**Summary Of Contributions:**

The paper introduces SCUT (Spectral Clustering for Unsupervised Classification Trees), a top-down hierarchical clustering method designed for tabular data. The key idea is to model the dataset as a bipartite graph connecting observations and features, then recursively split it by approximating the Normalized Cut objective using the SVD of a normalized weight matrix.

The authors emphasize three main contributions:
1) Density-based splitting: Instead of cutting the Fiedler vector at an arbitrary threshold (like zero), SCUT treats the 1D projection as a smooth distribution and looks for natural low-density gaps between peaks. Splits happen at these “valleys,” which tend to align better with meaningful structure in the data.
2) Out-of-sample prediction: At each internal node, SCUT learns and stores a linear classifier. This effectively turns the clustering procedure into an unsupervised decision tree, so new data points can be routed down the tree in logarithmic time—without recomputing the clustering from scratch.
3) Ante-hoc interpretability: By examining the feature-side singular vectors, the method measures how strongly each feature influences a split. This provides intuitive explanations for why the tree branches the way it does.

The method is evaluated on four varied datasets (Iris, French National Assembly voting, 20 Newsgroups, and Goodreads) and compared against Ward's method as well as an earlier bipartite spectral clustering approach by Zha et al. (2001).

**Audience:**

Yes

**Audience Explanation:**

The paper seems of possible interest to researchers and practitioners in the TMLR community, particularly those working on unsupervised learning, interpretability, and scalable methods for tabular data. Recasting hierarchical clustering as an “unsupervised decision tree” with built-in out-of-sample routing addresses a practical limitation of methods such as Ward's method, which are inherently transductive. The scalability analysis and demonstrations across multiple datasets might also suggest that the approach may be relevant across several application domains.

**Claims And Evidence:**

No

**Claims Explanation:**

While the quantitative evaluation would benefit from some revisions, the central claims of the paper, i.e., SCUT offers a scalable, divisive hierarchical clustering framework with built-in out-of-sample inference and ante-hoc interpretability, are largely supported by solid theory.

In particular:
1) The use of SVD on a bipartite graph to approximate the Normalized Cut objective is mathematically sound and builds naturally on prior spectral clustering work. In Appendix A.1, the authors go further and prove that the recursive splits induce strictly convex subspaces. That result is not just a technical detail, it provides real geometric justification for why the clusters are well-structured and why the out-of-sample routing mechanism (Algorithm 2) is principled rather than ad hoc.
2) The complexity discussion in Section 3.5 is detailed and technically sound. The derivation of worst-case $O(mn^2)$ and best-case $\Omega(mn\log n)$ bounds is formalized.
3) The method appears robust across data types, as the algorithm is applied to a diverse range of datasets.

**Requested Changes:**

1) It's not clear to me that out-of-sample inference is tested. Since a major selling point of SCUT is that it can place new data points into the learned hierarchy without rebuilding the tree (Algorithm 2), it would be good to have experiments demonstrating that it works well in practice. Currently, I believe all results are transductive, as clustering is performed on the full dataset, so the paper could consider including a train/test split experiment showing how reliably the learned linear classifiers assign unseen data, ideally compared against reclustering.

2) Similarly, the authors introduce a custom Top-Down (TD) metric, but they explicitly state that they searched for the best threshold $\theta$ to maximize accuracy on the same dataset used for evaluation. Hyperparameter tuning on the target data seems artificial and would further motivate a train/test split experiment in my opinion.

---

### Comment · Action_Editor_Egkn · 2026-04-16
**Main Points from Reviewer bJv7 regarding the Revision**

1) Claims.

1.1) The authors write: "However, most hierarchical clustering methods do not
support such updates and require the entire clustering to be recomputed-a
process that is computationally expensive and may result in significant
changes to the structure of the dendrogram." The reviewer claimed that this
generalization is not true, as the papers recommended by the reviewer --
(Cluster Tree for Nearest Neighbor Search, Kushnir et. al.), Wieling et al.
(2010, ACL Anthology), Zheng et al. (2018, Knowledge-Based Systems),
Sugahara et al. (2023, Pattern Recognition), Ienco, Pensa, Meo (ECML PKDD
2009) -- are also neither computationally expensive nor result in
significant changes of the structure of the dendrogram.

1.2) A claimed novelty of your work is that "the difference [to decision
tree inference] lies in the fact that the branching rules are learned from
spectral projections rather than from axis-aligned feature thresholds". The
reviewer claimed that this is not a novelty and pointed at above references.

1.3) The authors write: "While these methods leverage spectral principles or
hierarchical decompositions, they typically focus either on flat spectral
partitions, task-specific adaptations, or efficiency-oriented tree
structures, and generally lack a principled mechanism for inductive
assignment of new samples within a hierarchical framework." The reviewer is
concerned that this is again an overclaim, given the references above which
are quite similar.
--> There are hence several claims regarding the novelty of the approach
which are not supported, and which must be toned down.

2) Experimental evidence.

2.1) I agree with the reviewer that the experimental evaluation appears
insufficient, especially regarding the comparison with other approaches. As
I see it, SCUT is only compared to baselines, and not to the state of the
art. I would suggest considering the works cited above as methods for
comparison, or at least clarify why they cannot be used as baselines.

2.2) The reviewer pointed out that the results for the Iris dataset fall
behind the results of this method:
https://pmc.ncbi.nlm.nih.gov/articles/PMC7275956/pdf/CIN2020-1648573.pdf
--> While beating the state of the art is not essential for being accepted
at TMLR, some of the claims (even in the abstract: "We demonstrate, both
visually and quantitatively, that SCUT captures the intrinsic structure of
data more effectively than existing methods, while offering competitive
performance compared to common hierarchical clustering algorithms") require
a broader comparison to existing works.

---

### Decision · Action_Editor_Egkn · 2026-04-08

**Recommendation:** Reject

**Audience:**

Yes

**Audience Explanation:**

The reviewers agree that a part of the TMLR community may be interested in this work.

**Claims And Evidence:**

No

**Claims Explanation:**

This was a very difficult decision, with recommendations diverging substantially between the reviewers. The decision process was not taken lightly, and even had to be continued outside of OpenReview because some reviewers gave their recommendation before the revision was posted. While I acknowledge that the revision addresses many of the reviewers' comments and is generally well executed, two main concerns remain in regard of above question. The claims are not fully supported because:
1) Some claims in the text regarding the novelty of the manuscript are not substantiated relative to the existing body of literature.
2) The empirical evaluation seems incomplete, again referring to an existing body of literature that could -- possibly with adaptations -- at least act as better performing baselines.

With this in mind, at the moment I cannot accept the manuscript in its current form.

**Resubmission Of Major Revision:**

The authors may consider submitting a major revision at a later time.